


# Surface deposition of oxidized mercury dominated by production in the upper and middle troposphere

Viral Shah[1], Lyatt Jaeglé[1]

[1]Department of Atmospheric Sciences, University of Washington, Seattle, WA 98195

*Correspondence to*: Viral Shah (vshah@uw.edu)

**Abstract.** Oxidized mercury (Hg(II)) is chemically produced in the atmosphere by oxidation of elemental mercury and is directly emitted by anthropogenic activities. We use the GEOS-Chem global chemical transport model, with gaseous oxidation driven by Br atoms, to quantify how surface deposition of Hg(II) is influenced by Hg(II) production at different atmospheric heights. We tag Hg(II)

chemically produced in the lower (surface–750 hPa), middle (750–400 hPa) and upper troposphere (400 hPa–tropopause), in the stratosphere, as well as directly emitted Hg(II). A two-year simulation (2013–2014) reproduces the spatial distribution and seasonal cycle of Hg(II) surface concentrations and Hg wet deposition observed at the Atmospheric Mercury Network (AMNet) and the Mercury Deposition Network (MDN) stations over the United States to within 21%, but displays a 46% underestimate of

wet deposition observed at the European Monitoring and Evaluation Programme (EMEP) stations. We find that Hg(II) produced in the upper and middle troposphere constitutes 91% of the tropospheric mass of Hg(II) and 91% of the annual Hg(II) wet deposition flux. This large global influence from the upper and middle troposphere is the result of strong chemical production coupled with a long lifetime of Hg(II) in these regions. Annually, 77–84% of surface level Hg(II) over the western U.S., South

America, South Africa, and Australia is produced in the upper and middle troposphere, whereas 26–66% of surface Hg(II) over the eastern U.S., Europe, East Asia, and South Asia is directly emitted. Over the oceans, 72% of surface Hg(II) is produced in the lower troposphere, because of higher Br concentrations in the marine boundary layer. The global contribution of the upper and middle troposphere to the Hg(II) dry deposition flux is 52%. It is lower compared to the contribution to wet

deposition because dry deposition of Hg(II) produced aloft requires its entrainment into the boundary layer, while rain can scavenge Hg(II) from higher altitudes more readily. We find that 55% of the



spatial variation of Hg wet deposition flux observed at the MDN sites is explained by the combined variation of precipitation and Hg(II) produced in the upper and middle troposphere. Our simulation points to a large role of Hg(II) present in the dry subtropical subsidence regions, which account for 74% of Hg(II) at 500 hPa over the continental U.S., and more than 60% of the surface Hg(II) over high-altitude areas of the western U.S. During the Nitrogen, Oxidants, Mercury, and Aerosol Distributions, Sources, and Sinks (NOMADSS) aircraft campaign, the contribution of these dry regions was found to be 75% when measured Hg(II) exceeded 250 pg m$^{-3}$. Our results highlight the importance of the upper and middle troposphere as key regions for Hg(II) production and of the subtropical anticyclones as the primary conduits for the production and export of Hg(II) to the global atmosphere.

# 1 Introduction

Atmospheric deposition of mercury (Hg) is the main source of Hg to aquatic ecosystems. Methylmercury concentrations in fish in an ecosystem are strongly linked to the local Hg deposition rate (Hammerschmidt and Fitzgerald, 2006; Harris et al., 2007). Dry deposition and wet deposition are both significant contributors to the global deposition flux of Hg (e.g. Bergan et al., 1999; Seigneur et al., 2001; Dastoor and Larocque, 2004; Jung et al., 2009; Amos et al., 2012). While the global dry deposition fluxes of gaseous elemental mercury (Hg(0)) and oxidized mercury in the gas and particle phases (Hg(II)) are comparable, wet deposition occurs almost entirely through precipitation scavenging of Hg(II). Hg(II) is co-emitted with Hg(0) from several anthropogenic sources, but the predominant source of Hg(II) in the atmosphere is in situ oxidation of Hg(0) (Pirrone et al., 2010; Selin and Jacob, 2008; Holmes et al., 2010). Br is likely the main oxidant of Hg(0) (Ebinghaus et al., 2002; Laurier et al., 2003; Donohoue et al., 2006; Obrist et al., 2011; Gratz et al., 2015), but the importance of $O_3$ and OH is unclear (Hynes et al., 2009; Subir et al., 2011; Ariya et al., 2015).

Hg(II) concentrations in the planetary boundary layer are typically about 50 pg m$^{-3}$ (Valente et al., 2007; Gay et al., 2013), but could be as high as 1000 pg m$^{-3}$ in urban areas with large anthropogenic sources (Poissant et al., 2005; Fu et al., 2012) and in the Arctic during springtime Mercury Depletion Events (Cobbett et al., 2007). The free troposphere is thought to hold a global pool of elevated Hg(II) (Selin, 2009), but few Hg(II) observations have been made in the free troposphere. At high-elevation ground



sites, back-trajectory analysis and simultaneous measurements of $H_2O$ and $O_3$ were used to identify free-tropospheric air masses which contained higher Hg(II) concentrations compared to air masses transported from the planetary boundary layer (Swartzendruber et al., 2006; Faïn et al., 2009; Lyman and Gustin, 2009; Sheu et al., 2010; Timonen et al., 2013; Fu et al., 2016). Lyman and Jaffe (2012)

conducted aircraft-based measurements of Hg(II) in the upper troposphere and lower stratosphere and inferred that Hg(II) concentrations are highest at the tropopause. Other aircraft-based studies have also found increasing Hg(II) concentrations at 2-5 km altitude in the free troposphere (Sillman et al., 2007; Swartzendruber et al., 2009; Brooks et al., 2014). During the Nitrogen, Oxidants, Mercury, and Aerosol Distributions, Sources, and Sinks (NOMADSS) aircraft campaign, the highest Hg(II) concentrations

(300–680 pg m$^{-3}$) were observed in clean and dry air originating in the subsiding air masses of the Pacific and the Atlantic subtropical anticyclones (Gratz et al., 2015; Shah et al., 2016). Furthermore, higher concentrations of Hg in precipitation are observed in thunderstorms reaching higher altitudes (Guentzel et al., 2001; Shanley et al., 2015; Holmes et al., 2016), and higher Hg wet and dry deposition fluxes are associated with transport from the free troposphere (Weiss-Penzias et al., 2011; Gustin et al.,

2012; Huang and Gustin, 2012; Sheu and Lin, 2013).

The influence of free-tropospheric Hg on deposition has been evaluated with regional and global chemical transport models. Using the global GEOS-Chem model, Selin and Jacob (2008) estimated that 59% of the annual Hg(II) wet deposition over the U.S. is from Hg(II) scavenged from altitudes above 850 hPa. In another study (Myers et al., 2013), Hg present at the upper boundary (5.4 km) of the

regional CMAQ model was found to contribute about 40% to dry deposition and about 80% to wet deposition in July over the U.S. However, these model estimates are limited to specific regions and seasons.

In this study, we use the GEOS-Chem global chemical transport model to quantify the regional contributions of Hg(II) produced at different heights in the atmosphere to the annual deposition of

Hg(II). We have added a tagging method to the GEOS-Chem model to track Hg(II) produced in the lower (surface–750 hPa), middle (750–400 hPa) and upper troposphere (400 hPa–tropopause), Hg(II) produced in the stratosphere, and Hg(II) emitted by anthropogenic activities. This simulation is described and evaluated with ground-based observations of Hg(II) concentrations and wet deposition





(Sect. 2). In Sect. 3, we present the distribution of the tagged Hg(II) and calculate their contributions to wet and dry deposition fluxes in different regions of the world. We also examine the sensitivity of our results to different model assumptions for Hg chemistry and anthropogenic emission speciation. We use our simulation to examine the role of the subtropical anticyclones as global reservoirs of Hg(II)-rich air

(Sect. 4) and evaluate the role of tagged Hg(II) tracers in explaining the observed variability of Hg(II) concentrations and wet deposition fluxes (Sect. 5). Finally, we discuss the implications of our study in Sect. 6 and present conclusions in Sect. 7.

## 2 Observations and model used in this study

### 2.1 Observations of Hg wet deposition and atmospheric concentrations of Hg(II)

Hg wet deposition fluxes over North America and Europe are measured by the Mercury Deposition Network (MDN; http://nadp.sws.uiuc.edu/mdn/) and the European Monitoring and Evaluation Programme (EMEP; http://www.nilu.no/projects/ccc/index.html), respectively. These networks measure precipitation depth and Hg concentrations in precipitation weekly (MDN), biweekly (EMEP) or monthly (EMEP). In this study, we use the 2013–2014 monthly-mean and annual-mean wet deposition

flux and volume-weighted mean (VWM) Hg concentrations. The VWM concentration for any period is the total Hg wet deposition flux for that period divided by the total precipitation depth. All sites in the MDN network use standard instruments and protocols, and all samples are analyzed at the same laboratory (Prestbo and Gay, 2009). The measurement precision in MDN observations, estimated from collocated sampling, is less than 15% (Wetherbee et al., 2007). A field inter-comparison of instruments

and methods used in the EMEP network found the measurement precision for the EMEP network to be about 40% (Aas, 2006).

To calculate monthly means, we discard sites with fewer than 3 weeks of measurements in any given month. For annual means we require at least 8 months of valid measurements. The MDN network had 80 stations over the continental U.S. that met the above data completeness criteria during 2013–2014,

whereas the EMEP network had 9 stations over Europe.





The Atmospheric Mercury Network (AMNet; http://nadp.sws.uiuc.edu/amn/) monitors surface concentrations of Hg(0), reactive gaseous mercury (RGM) and particle-bound mercury (PBM). The sum of RGM and PBM is considered to represent Hg(II). RGM and PBM measurements are made on a 2- or 3-hour cycle, depending on the site. All AMNet stations use the Tekran® 2537-1130-1135 speciation system, and follow operational procedures described in Gay et al. (2013). There is no standard calibration method for Tekran RGM and PBM measurements, and the uncertainties in these measurements are not fully quantified. A few studies have found that the AMNet instruments underestimate RGM by a factor of 2–3 in the presence of ambient water vapor and $O_3$ (Lyman et al., 2010; Ambrose et al., 2013; McClure et al., 2014). Here, we use the 2009–2012 AMNet observations, as this data is publicly available. AMNet had 23 sites over the continental U.S. and eastern Canada (Nova Scotia) operational during this period. The annual and monthly statistics for each station are calculated by aggregating 2- or 3-hour measurements made during 2009–2012.

The NOMADSS aircraft campaign took place over the eastern U.S. from June 1 to July 15, 2013. Total Hg and Hg(II) observations were made with the University of Washington Detector of Oxidized Hg Species (DOHGS) instrument (Ambrose et al., 2015; Swartzendruber et al., 2009; Lyman and Jaffe, 2012). The detection limit of the DOHGS instrument for Hg(II) measurements during the campaign was between 57 and 228 pg m$^{-3}$, and we use the robust Regression on Order Statistics (ROS) to estimate values for measurements below detection limit, as described by Shah et al. (2016).

## 2.2 GEOS-Chem model

GEOS-Chem is a global chemical transport model that simulates the emissions, transport, chemistry, and deposition of Hg(0), gas-phase Hg(II), and particle-phase Hg(II) (Selin et al., 2007). The model is driven by meteorological fields from the NASA Global Modeling and Assimilation Office (GMAO) Goddard Earth Observing System Model, Version 5 FP (GEOS-5 FP) modeling system. The GEOS-5 FP system consists of a general circulation model coupled with a data assimilation system (Reinecker et al., 2008), and has a native horizontal resolution of 0.25° latitude × 0.3125° longitude with 72 vertical levels up to 0.01 hPa. We average the meteorological fields to a coarser resolution of 2° latitude × 2.5° longitude and 47 vertical levels for the GEOS-Chem simulations in this study. Global anthropogenic



emissions of Hg are from the global United Nations Environment Programme (UNEP) / Arctic Monitoring and Assessment Programme (AMAP) 2010 inventory (http://www.amap.no/mercury-emissions/datasets). We assume that stack emissions (emission height > 50m) of Hg consist of 90% Hg(0) and 10% Hg(II). Natural emissions are simulated using a slab ocean model (Strode et al., 2007;

5    Soerensen et al., 2010) and a land emissions model (Selin et al., 2008). Emissions from biomass burning and geogenic activity are prescribed as in Holmes et al. (2010). Transport processes simulated in the GEOS-Chem model include advection (Lin and Rood, 1996), convective transport (Wu et al., 2007), and turbulent mixing in the boundary layer (Lin and McElroy, 2010).

The redox chemistry of Hg consists of oxidation of Hg(0) by Br, as described below, and aqueous phase

10   reduction in the presence of sunlight (Holmes et al., 2010). Gas / particle partitioning of Hg(II) on sea-salt aerosols is simulated as a kinetic process (Holmes et al., 2010), while partitioning on other aerosols is simulated as an equilibrium process (Amos et al., 2012). The oxidation of Hg(0) to Hg(II) is simulated as follows (Goodsite et al., 2004; Balabanov et al., 2005; Dibble et al., 2012):

$$Hg(0) + Br \xleftarrow{\phantom{xx}M\phantom{xx}} HgBr \ 15 \tag{R1}$$
$$HgBr + Br \longrightarrow Hg(0) + Br_2 \tag{R2}$$
$$HgBr + X \longrightarrow Hg(II) \tag{R3}$$
$$(X = NO_2, HO_2, BrO, Br, OH)$$

with the following reaction rates:

$$k_{1f} = 1.46 \times \left(\frac{T}{298}\right)^{-1.86} \times [M] \quad cm^3 \ molecule^{-1} \ s^{-1}$$

$$k_{1r} = 2.67 \times 10^{41} \times \exp\left(\frac{-7292}{T}\right) \times \left(\frac{T}{298}\right)^{1.76} \times k_{1f} \quad s^{-1}$$

$$k_2 = 3.9 \times 10^{-11} \quad cm^3 \ molecule^{-1} \ s^{-1}$$

$$k_3 = 2.5 \times 10^{-10} \times \left(\frac{T}{298}\right)^{-0.57} \quad cm^3 \ molecule^{-1} \ s^{-1}$$





Concentrations of Br, BrO, $NO_2$, $HO_2$, and OH are obtained from the archived monthly-mean output of the 4° latitude × 5° longitude $HO_x$-$NO_x$-$O_3$-VOC-Br GEOS-Chem simulation for 2013 (Bey et al., 2001; Parrella et al., 2012). In our previous work (Shah et al., 2016), we found that the GEOS-Chem Br concentrations simulated by Parrella et al. (2012) were insufficient in explaining Hg(II) concentrations

observed during the NOMADSS aircraft campaign at 5–7 km altitude. We found improved agreement with NOMADSS Hg(II) observations when we increased Br concentrations by a factor of 3 between 45°S and 45°N and between 750 hPa and the tropopause. Schmidt et al. (2016) have recently updated the GEOS-Chem bromine simulation by expanding the multiphase chemistry of bromine to include reactions with chlorine and ozone. These updates result in faster recycling of HBr to $BrO_x$ and a factor

of 2.5 increase in tropospheric Br concentrations for 45°S–45°N above 2.5 km, improving agreement with satellite and in situ observations of BrO. This is consistent with our assumption that Br concentrations are 3 times higher than those simulated with the previous mechanism. In addition, these updates by Schmidt et al. (2016) have resulted in a factor of 2.3 increase in free tropospheric Br concentrations at higher latitudes (45°N–90°N). To maintain consistency with our previous work, we

continue to use the Parrella et al. (2012) Br fields with the factor of 3 scaling in this study too, but note that Br concentrations north of 45°N may be too low.

The GEOS-Chem model includes wet deposition of Hg(II) and dry deposition of Hg(0) and Hg(II). Wet deposition includes in-cloud scavenging (rainout) and below-cloud scavenging (washout) in convective and large-scale precipitation (Liu et al., 2001). Within clouds, the dissolution of gas-phase Hg(II) in

liquid droplets is modeled as an equilibrium process, while particle-phase Hg(II) is assumed to be fully dissolved (Amos et al., 2012). We assume that rainout of gas-phase Hg(II) does not occur during ice nucleation (T < 248 K). Below clouds, gas-phase Hg(II) is washed out by dissolving in falling raindrops (T > 268K), but not in falling snow and ice (Amos et al., 2012). Particle-phase Hg(II) is washed out in collisions in falling rain, snow and ice with different efficiencies (Wang et al., 2011). Dry deposition of

gas-phase Hg(II) and particle-phase Hg(II) on particles other than sea-salt aerosols is based on the resistance-in-series model (Wesely, 1989; Wang et al., 1998; Zhang et al., 2001). The surface resistance of gas-phase Hg(II) is assumed to be negligibly small (Selin et al., 2007; Amos et al., 2012). The dry deposition of particle-phase Hg(II) present on sea-salt aerosols is parameterized using results of a box



model simulating the chemistry and deposition of Hg(II) in the marine boundary layer (Holmes et al., 2009, 2010).

### 2.2.1 Model uncertainties

Uncertainties in mercury modeling and chemistry have been recently reviewed by Gustin et al. (2015),
Ariya et al. (2015), and Kwon and Selin (2016). Here we briefly discuss uncertainties which are pertinent to our study: uncertainties in the assumption of Br as the sole oxidant of Hg(0), in reduction kinetics of Hg(II), and in the assumed speciation of Hg(0) and Hg(II) in anthropogenic emissions.

While Br, $O_3$, and OH have been identified as possibly important oxidants of Hg(0), there is growing evidence from theoretical (Goodsite et al., 2004; Dibble et al., 2012), laboratory (Ariya et al., 2002; Donohoue et al., 2006) and field studies (Ebinghaus et al., 2002; Lindberg et al., 2002; Laurier et al., 2003; Obrist et al., 2011; Gratz et al., 2015) that Br may be the most relevant oxidant of Hg(0) in the atmosphere. Ab-initio calculations have suggested that HgO, the product of gas-phase oxidation of Hg(0) by $O_3$ and OH, is a weakly-bound molecule, and that oxidation of Hg(0) by $O_3$ and OH is an endothermic reaction of little importance in the atmosphere (Hynes et al., 2009).

The pathways for reduction of Hg(II) to Hg(0) in the atmosphere are poorly characterized. Laboratory experiments suggest that photoreduction of Hg(II) can occur in the aqueous-phase in the presence of organic compounds or on dry aerosol surfaces at atmospherically relevant rates (Si and Ariya, 2008; Tong et al., 2013), and field studies have found some evidence for in situ reduction of Hg(II) (Edgerton et al., 2006; Landis et al., 2014; de Foy et al., 2016). Most global atmospheric mercury models include at least one pathway of Hg(II) reduction in order to simulate realistic Hg(0) concentrations (Ariya et al., 2015). The reduction rate of aqueous-phase Hg(II) in GEOS-Chem is parameterized based on the simulated $NO_2$ photolysis rate (Holmes et al., 2010). We adjust the reduction rate to best match aircraft- and ground-based observations of Hg(0) over the mid-latitudes.

We have assumed an emissions speciation of 90% Hg(0) and 10% Hg(II) for anthropogenic emissions from stacks, as opposed to the UNEP/AMAP speciation of 55% Hg(0) : 45% Hg(II) for stack sources. Zhang et al. (2012) and Kos et al. (2013) found that a speciation scheme with 10–15% of Hg(II), and the rest Hg(0), best explains the spatial variability in Hg(II) wet deposition observed over the U.S.



However, the speciation of Hg emissions can vary considerably based on the type of source, type of pollution control devices, and the availability of oxidants in the flue gas (Kim et al., 2010).

### 2.2.2 Simulations performed for this study

We have added a tagging technique to the GEOS-Chem model to identify the production regions of Hg(II). We divide the atmosphere vertically into lower troposphere (LT: surface–750 hPa), middle troposphere (MT: 750–400 hPa), upper troposphere (UT: 400 hPa–tropopause) and stratosphere (STRAT), to track the Hg(II) produced in each of these regions as separate Hg(II) tracers. Hg(II) emitted directly to the atmosphere is also tagged separately (E–Hg(II)). Each of these tagged tracers undergo the same physical and chemical processes as the total Hg(II) tracer. Hg(II) loss by deposition or reduction in a model grid cell is divided among all tagged tracers present in the grid cell in proportion to their masses. We perform a simulation with the tagged tracers for the years 2013 and 2014 following a model spin-up period of six years.

We perform an additional simulation to quantify the role of the dry subsidence regions of the subtropical anticyclones in the global transport of Hg(II). We identify the dry subtropical subsidence areas as those that lie between 45°S and 45°N and between 750 hPa and the tropopause and where the monthly-mean relative humidity is less than 20%. The relative humidity threshold is based on the definition of dry subtropical areas of Cau et al. (2007). We introduce duplicate Hg(II) tracers that are produced and lost exactly as the original Hg(II) tracers, but at each time step we set to zero the concentrations of these tracers within the dry subtropical areas. The amount of Hg(II) originating in the dry areas (dry-Hg(II)) is then calculated by difference between the original and the duplicate Hg(II) tracers. This simulation is performed for the year 2013.

In addition, we perform three one-year (2013) sensitivity simulations with the tagged tracers addressing uncertainties in mercury oxidation and Hg(0):Hg(II) partitioning in anthropogenic emissions (Sect. 2.2.1).





### 2.2.3 Comparison of modeled and measured Hg(II)

Figures 1–3 compare the modeled Hg(II) concentrations and wet deposition fluxes to observations from the MDN, EMEP, and AMNet networks. The modeled annual wet deposition flux at the MDN sites (10.4 ± 4.2 µg m$^{-2}$ a$^{-1}$; mean ± standard deviation) is in close agreement with observations (10.2 ± 4.0 µg m$^{-2}$ a$^{-1}$) (Fig. 1a). The model reproduces the observed spatial pattern in annual wet deposition fluxes ($r^2$=0.67). Wet deposition is lowest in western U.S. (MDN: 6.9 µg m$^{-2}$ a$^{-1}$, GEOS-Chem: 6.2 µg m$^{-2}$ a$^{-1}$), higher in the northeast U.S. (MDN: 8.5 µg m$^{-2}$ a$^{-1}$, GEOS-Chem: 8.4 µg m$^{-2}$ a$^{-1}$) and in the central U.S. (MDN: 11.2 µg m$^{-2}$ a$^{-1}$, GEOS-Chem: 13.2 µg m$^{-2}$ a$^{-1}$), and largest in the southeast U.S. (MDN: 15.4 µg m$^{-2}$ a$^{-1}$, GEOS-Chem: 15.2 µg m$^{-2}$ a$^{-1}$). The observed monthly-mean wet deposition fluxes exhibit a seasonal maximum in summer, particularly in the central, northeast and southeast regions (Fig. 1c). This seasonality is driven by an increase in precipitation and an increase in mercury concentrations in precipitation (Prestbo and Gay, 2009; Selin and Jacob, 2008). The model reproduces the observed seasonal variations in the central and northeast regions, but underestimates the summer deposition fluxes in the southeast because of a factor of 2 underestimate in summertime precipitation by the GEOS-5 FP meteorological fields (not shown). Overall, 66–88% of the modeled wet deposition fluxes are within a factor of 2 of the observations (FAC2; FAC2=fraction of points where $0.5 \leq M_i / O_i \leq 2$ where $O_i$ and $M_i$ are observed and simulated values, respectively) for the four regions, and the normalized mean bias (NMB; NMB=$\sum_i \left( M_i - O_i \right) \Big/ \sum_i O_i$ ) ranges between -7% and +20%.

The model also captures the observed annual VWM concentrations (MDN: 10.0 ± 4.3 ng L$^{-1}$; GEOS-Chem: 9.7 ± 4.7 ng L$^{-1}$) (Fig. 1d). Higher VWM concentrations are observed in the western and central U.S. (11.6 and 14.1 ng L$^{-1}$ respectively) compared to the northeast and southeast (7.9 and 10.6 ng L$^{-1}$ respectively), indicating the presence of higher atmospheric concentrations of Hg(II) over these regions. Modeled VWM concentrations show a spatial pattern similar to observations: higher values in western (8.7 ng L$^{-1}$) and central (14.4 ng L$^{-1}$) U.S. and lower values in northeast (6.7 ng L$^{-1}$) and southeast (13.6 ng L$^{-1}$). In western and central U.S. the observed and modeled VWM concentrations show a pronounced summer maximum (Fig. 1d), while in northeast and southeast the seasonal cycle is weaker. We find that 65–96% of the modeled monthly VWM concentrations are within a factor of 2 of the observations, with



a NMB ranging between -22% and +33%. Over southeast U.S., tmodeled VWM concentrations are higher than observations during winter and spring, suggesting a model overestimate in atmospheric Hg(II) concentrations in that region, or an overestimate in the amount of Hg(II) scavenged by precipitation.

Over Europe (Fig. 2a), the modeled wet deposition flux ($3.5 \pm 1.4$ µg m$^{-2}$ a$^{-1}$) underestimates observations at EMEP sites ($6.1 \pm 3.1$ µg m$^{-2}$ a$^{-1}$). Similarly, modeled VWM concentrations ($3.6 \pm 1.0$ ng L$^{-1}$) are significantly lower than observations ($6.0 \pm 1.8$ ng L$^{-1}$) (Fig. 2b). The summertime underestimate is partially explained by a 40% underestimate of observed summertime precipitation by the GEOS-5 FP meteorological fields, but the discrepancy exists year-round. The remaining

discrepancy could indicate an underestimate in the modeled Hg(II) concentrations over the region, likely because the upward scaling of the Br concentrations in our simulation did not extend north of 45°N and covered only parts of southern Europe (Section 2.2). The modeled seasonal cycle in wet deposition shows higher fluxes from April to August, following qualitatively the observed seasonal cycle (Fig. 2c). We find that 55% of the modeled monthly-mean wet deposition fluxes are within a

factor of 2 of the observations, with a NMB of -46%. Model and observations display a similar seasonal cycle in VWM, with higher concentrations in April through August (Fig. 2d). The FAC2 and NMB statistics for the modeled VWM concentrations are 63% and -41% respectively, suggesting that the modeled oxidation rate is too slow over this region.

In Fig. 3 we present a comparison of modeled surface Hg(II) concentrations with observations at

AMNet sites. Modeled Hg(II) surface concentrations ($11.7 \pm 8.3$ pg m$^{-3}$) are comparable to observations ($15.0 \pm 8.2$ pg m$^{-3}$) (Fig. 3a). The model simulates enhanced Hg(II) surface concentrations (25–40 pg m$^{-3}$) over the intermountain region of the western U.S., consistent with AMNet observations in Utah. During summer, observed and modeled Hg(II) concentrations reach a minimum in the eastern U.S. (Fig. 3b). This is due to multiple factors: larger losses of Hg(II) by wet deposition and reduction induced by

increasing low cloud coverage and precipitation, as well as decrease in Hg(II) production following the seasonal cycle in Hg(0) concentrations (Amos et al., 2012; Zhang et al., 2012). Overall, we find that 70% of monthly-mean modeled concentrations are within a factor of 2 of AMNet observations in the





eastern U.S., with a NMB of -21%. If we assume that the reported RGM is underestimated by factor of 3 due to interferences (see Sect. 2.1), we find a model NMB of -57%.

## 3 Tagged simulation results

### 3.1 Global distribution of tagged Hg(II) tracers

The annual zonal mean distribution of modeled Hg(II) concentrations is shown in Fig. 4a. Hg(II) concentrations increase from 10 pg m$^{-3}$ near the surface to 1000 pg m$^{-3}$ in the upper troposphere, and exhibit local maxima in the subtropical middle troposphere, within the descending Hadley branches. The chemical production rate of Hg(II) (via reactions R1–R3, Fig. 4b) increases by an order of magnitude between the lower and upper troposphere. This increase is driven by increasing Br concentrations coupled with colder temperatures (hence slower thermal decomposition of HgBr in R1) (Holmes et al., 2010). Regions of high Hg(II) production rates also occur near the surface in the Arctic due to springtime release of Br in bromine explosion events (Holmes et al., 2010). The elevated Southern Ocean production rates are associated with high emissions of sea-salt aerosol, which are assumed to release bromine (Parrella et al., 2012). Note that the sharp gradients in Hg(II) production rates at 45°N and 45°S reflect the boundaries of the Br scaling in the model.

The lifetime of Hg(II) increases from less than 1 day in the lower troposphere to over 3 years in the tropical upper troposphere. Hg(II) in the lower troposphere is subject to dry deposition, and in-cloud reduction and scavenging by precipitation in the lower and middle troposphere. Thus, despite higher production rates, Hg(II) concentrations over the Arctic and the Southern Ocean are low. The long lifetimes of Hg(II) in the upper troposphere and in the descending branches of the Hadley circulation are due to infrequent occurrence of reduction within clouds and wet scavenging. As summarized in Table 1, we find that the lifetime of Hg(II) is highest for the STRAT tracer (45 days), and decreases to 22 days, 4.1 days, and 0.6 days for the UT, MT, and LT tracers, respectively.

The large production rates of Hg(II) in the upper and middle troposphere combined with a longer lifetime result in the large contributions of the UT and MT tracers to the tropospheric mass and deposition of Hg(II). Overall, the tropospheric burden of Hg(II) (616 Mg) is dominated by Hg(II)




produced in the UT (514 Mg, 83%), with smaller contributions from the stratosphere (STRAT: 50 Mg, 8%) and the MT (47 Mg, 8%), and less than 1% from the LT (4 Mg) and direct emissions (1 Mg) (Table 1). The UT tracer contributes 67% and 35% of the Hg(II) burden in the middle and lower troposphere respectively (Table 1 and Fig. 4d), and the MT tracer accounts for 40% of Hg(II) in the lower

troposphere (Table 1 and Fig. 4e). The LT tracer accounts for 20% of the Hg(II) burden in the lower troposphere (Table 1), and its contribution increases to >50% near the surface over the Arctic and the Southern Ocean (Fig. 4f). We also find that 70% of Hg(II) in the lowermost stratosphere is comprised of the UT tracer(Fig. 4d), because Hg(0) is rapidly oxidized in the upper troposphere and is almost completely depleted before reaching the stratosphere, as shown by observations (Talbot et al., 2007;

Lyman and Jaffe, 2012). The E-Hg(II) tracer accounts for 5% of the Hg(II) burden in the lower troposphere (Table 1), but its contribution increases to >10% over the northern mid-latitudes (Fig. 4h). We find that 54% of the global production of Hg(II) occurs in the UT (8560 Mg a$^{-1}$), with smaller contributions from the MT (27%), LT (16%), STRAT (3%), and direct emissions (1%) (Table 1). Together, the UT and MT tracers account of 91% of global surface wet deposition (60% from UT and

31% from MT) and 52% of dry deposition (24% from UT and 28% from MT). Their higher contributions to wet deposition is because precipitation scavenging can directly remove these tracers from higher altitudes, while dry deposition requires the transport of these tracers to the planetary boundary layer.

### 3.2 Origin of Hg(II) in surface deposition and concentrations

As shown in Fig. 5a, the highest surface Hg(II) concentrations (>50 pg m$^{-3}$) are simulated over high-elevation areas (e.g., western U.S., the Andes, and the Tibetan plateau), in polar regions, near emission sources (e.g., East Asia), and in dry subtropical areas (e.g., the Sahara desert, southern Africa, and Australia). Modeled Hg(II) concentrations are generally low over the oceans because of fast removal by sea-salt aerosols. Together, the UT and MT tracers account for 63% of surface Hg(II) over the

continents (Fig. 5e and 6b). Hg(II) over most of the oceans is predominantly from the LT tracer (Fig. 5e). In the subtropical anticyclones, free-tropospheric air is entrained in the marine boundary layer due to large-scale subsidence causing higher contributions from the MT. For western U.S., South America,





Africa, and Australia, UT and MT each make a contribution of about 40% to surface Hg(II), whereas for the Pacific and the North Atlantic Oceans 57% of surface Hg(II) is from the LT tracer (Fig. 6b). The contribution of E-Hg(II) to surface Hg(II) concentrations is limited to regions with high anthropogenic emissions. We calculate that 27–69% of surface Hg(II) in eastern U.S., Europe, East and South Asia

consists of E-Hg(II) (Fig. 6b).

The global distribution of the Hg(II) wet deposition flux (Fig. 5b) largely follows the spatial distribution of precipitation, with high wet deposition along the Intertropical Convergence Zone (ITCZ) and in the mid-latitude storm tracks. Globally, the UT tracer accounts for 60% of Hg(II) in wet deposition, but in some areas over South America, Africa, and Asia it exceeds 70% (Fig. 5f). The MT tracer makes up

most of the remaining fraction of wet deposition, with a global average contribution of 31% (Table 1). The contribution from the LT tracer is significant only at high latitudes, while the contribution from E-Hg(II) is noticeable mainly in East Asia. The relative wet deposition contributions of the tagged Hg(II) tracers remain fairly uniform across the ten regions summarized in Fig. 6c.

The Hg(II) dry deposition flux (Fig. 5c) maximizes in the subtropical anticyclones, where subsidence

provides a source of free-tropospheric Hg(II) to the planetary boundary layer. In addition, local maxima occur downwind of the emissions regions of the eastern US and East Asia, over high-elevation regions in western U.S. and the Himalayas, and over the Southern Ocean. In terms of the tagged tracers, their spatial contribution to dry deposition (Fig. 5g) is similar to their contribution to surface Hg(II) concentrations (Fig. 5e). We find that 79–82% of the Hg(II) dry deposition over western U.S., South

America, Africa, and Australia is from the UT and MT tracers. The E-Hg(II) tracer contributes 21–62% to dry deposition over eastern U.S., Europe and South and East Asia (Fig. 6d). Over the Pacific and North Atlantic Oceans, the UT, MT, and LT tracers each contribute about 30% to the dry deposition flux (Fig. 6d).

Our estimate that 92% of Hg(II) wet deposition and 73% of dry deposition over the U.S. is contributed

by production in the upper and middle troposphere is qualitatively consistent with the estimates of Selin and Jacob (2008). They calculated that 59% of the Hg(II) wet deposited over the U.S. was scavenged above 1.5 km, and that 70% of the Hg(II) below 1.5 km was transported from elsewhere. For comparison, with our simulation we find that 85% of the Hg(II) wet deposited over the U.S. is



scavenged above 1.5 km (note that to be consistent with Selin and Jacob (2008), we are comparing here the contribution of Hg(II) *present* above 1.5 km, and not the Hg(II) *produced* above 1.5 km). While Selin and Jacob (2008) also used the GEOS-Chem model, their simulation was based on Hg(0) oxidation by OH and $O_3$, while ours is based on oxidation by Br. In Sect. 3.3, we quantify the sensitivity of our results to the oxidation pathway assumed.

**3.3 Model sensitivity to oxidation chemistry and emission speciation**

We now assess the sensitivity of our results to our assumptions about mercury oxidation and Hg(0):Hg(II) partitioning in anthropogenic emissions. We perform three additional one-year (2013) sensitivity simulations with the following changes with respect to the base simulation: (i) use of the original GEOS-Chem Br concentrations instead of the 3 times Br concentrations in the base simulation, (ii) use of $O_3$ and OH as the Hg(0) oxidants (rate constants of Hall (1995) and Sommar et al. (2001)) instead of Br, and (iii) with the default UNEP/AMAP Hg(0):Hg(II) emission speciation of 55%:45% instead of the modified speciation.

When we use the original GEOS-Chem Br concentration, the contribution of the UT and MT tracers to the tropospheric Hg(II) burden decreases from 91% to 78%. The contribution of the STRAT tracer increases to 21% compared to 8% in the base simulation. The contribution of UT and MT tracers to total deposition (64%) is also smaller compared to the base simulation (76%), while the contribution of LT tracer to total deposition increases from 20% to 27%. In the $O_3$ and OH oxidation simulation, the contribution of the MT (18%) and LT (4%) tracers to the tropospheric Hg(II) mass is higher compared to the base simulation (MT: 8%, LT: <1%), while the contribution of the UT tracer decreases to from 83% to 60%. For total deposition, a larger fraction of Hg(II) production occurs in the middle and lower troposphere with the $O_3$ and OH oxidation simulation: LT: 38%, MT: 38%, UT: 19% (base, LT: 20%, MT: 30%, UT: 46%). When we change the Hg(0):Hg(II) emission speciation, the contribution of E-Hg(II) tracer to total deposition increases to 4% from 2% in the base simulation, but the contribution of the UT and MT tracers to deposition remains nearly unchanged. In all three sensitivity simulations, we find that the UT+MT tracers together contribute significantly to the tropospheric mass (78–90%) and the surface deposition flux (57–76%) of Hg(II), thus our overall conclusions remain robust.



## 4 Role of the subtropical dry regions

In this section, we focus on the specific role of subtropical anticyclones as a global reservoir of Hg(II). We quantify the Hg(II) that is transported from the subtropical anticyclones (dry-Hg(II) tracer) with a simulation where we artificially set to zero the Hg(II) present in the subtropical dry areas, which we define as RH < 20% and latitude < 45° (Sect. 2.2.2). Figure 7 shows the contribution of dry-Hg(II) to the annual-mean surface Hg(II), 500 hPa Hg(II), and Hg(II) wet deposition. Areas at 500 hPa where RH was less than 20% for minimum of four months in the year are shown in Fig. 7b. Based on our definition, we find that the dry areas contain 8% of the tropospheric mass of air.

We see from Fig. 7a that Hg(II) present in the subtropical dry areas exerts a strong influence on Hg(II) concentrations at the surface between 40°S and 40°N. The influence is stronger over the continents than over the oceans, where the local production of Hg(II) in the marine boundary layer is larger. More than 80% of the surface Hg(II) over dry areas in Africa, the Middle East, and Australia, and in the high-elevation regions of western U.S., Tibetan Plateau, and South America consists of dry-Hg(II). The influence of dry-Hg(II) on surface Hg(II) concentrations is small in anthropogenic Hg(II) source regions such as eastern U.S. and East Asia, and in regions that experience deep convection such as the ITCZ in the Pacific, Atlantic, and Indian Oceans, South Asia, the Maritime Continent. Surface Hg(II) in areas poleward of 40° is from anthropogenic emissions (Europe), is produced locally (polar regions), or is transported from higher altitudes in transient large-scale eddies (Canada and Russia).

The bulk of the mass of Hg(II) present at 500 hPa in the 40°S–40°N band is made up of dry-Hg(II) (Fig. 7b). The contribution of dry-Hg(II) extends far beyond the boundaries of the dry areas, suggesting that these regions act as global suppliers of Hg(II). Our model simulation suggests that 74% of the Hg(II) present at 500 hPa over the continental U.S. is transported from the dry subsidence band over the Pacific Ocean. The contribution of dry-Hg(II) decreases north of 40°N, but is still larger than 25% over most parts of Canada, Europe, and northern Asia. The contribution of dry-Hg(II) to Hg(II) wet deposition falls in-between the contributions of dry-Hg(II) to the surface and 500 hPa concentrations (Fig. 7c), as most precipitation scavenging of Hg(II) occurs between the surface and 500 hPa. In areas of the globe with large deposition fluxes (the ITCZ and the mid-latitude storm tracks) at least 50% of



the deposition consists of dry-Hg(II). Globally, dry-Hg(II) accounts for 74% to the tropospheric Hg(II) mass, and 59% of the total Hg(II) deposition (wet: 62% and dry: 52%).

During the 2013 NOMADSS aircraft campaign, high Hg(II) concentrations were observed and simulated above 5 km altitude (observations: $189 \pm 103$ pg m$^{-3}$; model: $165 \pm 104$ pg m$^{-3}$) (Shah et al., 2016). Back trajectory calculations indicated that these air masses were transported from even higher altitudes within the Pacific and the Atlantic anticyclones (Gratz et al., 2015; Shah et al., 2016). We sample the GEOS-Chem model along the NOMADSS flight tracks to determine the contribution of dry-Hg(II) to the Hg(II) concentrations measured during the campaign. We find that dry-Hg(II) accounted for 75% of Hg(II) when observed Hg(II) concentrations exceeded 250 pg m$^{-3}$ (Fig. 8). The dry-Hg(II) contribution decreased for observations with lower Hg(II) concentrations: 58% for 200–250 pg m$^{-3}$ and 10–20% for concentrations below 200 pg m$^{-3}$. The association between NOMADSS observations of high Hg(II) concentrations and higher contribution of dry-Hg(II) adds support to our finding that the subsidence regions act as a large source of Hg(II) present and deposited over the U.S.

Our finding is consistent with ground-based Hg(II) observations in western U.S., an area heavily influenced (>60%) by Hg(II) present in the dry subtropical regions (Fig. 7). Weiss-Penzias et al. (2009) reported that occurrence of higher (~50 pg m$^{-3}$) RGM concentrations in Nevada during June–August were associated with subsiding air in the anticyclone located over the Pacific Ocean, and Huang and Gustin (2012) found higher than mean Hg(II) deposition in Nevada under similar patterns of air transport. Timonen et al. (2013) showed that the highest concentrations of RGM (700 pg m$^{-3}$) observed at the Mt. Bachelor Observatory in Oregon (2.7 km altitude) corresponded to air masses transported from the subtropical Pacific Ocean.

## 5 Tagged tracer contributions at MDN and AMNet sites

Our tagged simulation show that the upper and middle troposphere are the predominant regions of production of Hg(II). Thus, areas where wet deposition is strongly influenced by Hg(II) produced in these regions can be expected to have higher wet deposition flux of Hg(II). We now examine whether such an enhancement in Hg wet deposition flux is indeed observed at MDN sites. Figure 9 shows the relationship between observed MDN annual Hg wet deposition fluxes to precipitation and modeled



contribution of the UT and MT tracers to the wet deposition flux at the site locations. As expected, we see that Hg wet deposition fluxes increase with increasing precipitation (e.g. Prestbo and Gay, 2009; Selin and Jacob, 2008). In addition, we find that the Hg wet deposition fluxes increase with increasing contribution of the UT and MT tracers to the wet deposition flux. Using multiple linear regression, we

derive the following relationship between Hg flux, precipitation amount, and contribution of UT+MT: $\text{Flux} = 0.004 \times (\text{Precipitation}) + 0.8 \times (\text{UT+MT contribution}) - 68$ . The regression parameters are statistically significant (p<0.001, 2-sided t-test, N=80), implying that both higher precipitation amounts and higher contribution of UT+MT tracers to wet deposition result in higher Hg flux. Precipitation amounts and the contribution of UT and MT together explain 55% of the spatial variation in the

observed Hg flux, while individually they explain 25% and 42% of the spatial variation, respectively. AMNet sites in the eastern U.S. are close to regional Hg(II) emission sources, and are thus more likely to be influenced by Hg(II) directly emitted rather than by Hg(II) produced aloft. Figure 10 shows that the 2009–2012 median Hg(II) concentrations observed at the AMNet sites in the eastern U.S. are higher at sites where the contribution of E-Hg(II) tracer is higher. For example, the surface Hg(II)

concentrations at sites NY06, WV99, and MD08 are ~10 pg m$^{-3}$, with 60–65% of the Hg(II) due to the E-Hg(II) tracer. On the other hand, at the remote site NS01, Hg(II) concentration are 3 pg m$^{-3}$ and the contribution of E-Hg(II) tracer is less than 10%. We find that spatial variation in the contribution of the E-Hg(II) tracer explains 27% of the variation in observed surface Hg(II) concentrations at the AMNet sites (excluding the outlier NY95) in the eastern U.S. (Fig. 10b). A statistically significant linear

relationship (p=0.018, 2-sided t-test, N=20, NY95 excluded) between Hg(II) concentrations and the contribution of the E-Hg(II) tracer is obtained from ordinary least squares regression. This suggests that although Hg(II) produced in the free troposphere makes up a large part of Hg(II) in the planetary boundary layer, spatial variations in Hg(II) concentrations in areas close to Hg(II) sources reflect variations in the amount of directly emitted Hg(II).

**6 Implications**

Our modeling study indicates that 91% of the global mercury wet deposition flux consists of Hg(II) produced in the upper and middle troposphere. Even in areas with large anthropogenic sources of





Hg(II), such as Europe and East Asia, directly emitted Hg(II) makes up less than 30% of the regional wet deposition flux, while Hg(II) produced locally in the lower troposphere accounts for less than 5%. This implies that regional decreases in anthropogenic Hg emissions do not lead to a proportional regional decrease in wet deposition. Indeed, numerous studies have demonstrated the importance of

intercontinental transport to mercury wet deposition (see Pirrone and Keating, 2010 and references therein). For example, Jaeglé et al. (2009) found that a 20% decrease in regional anthropogenic mercury emissions in the GEOS-Chem model leads to between 3% (North America) and 12% (East Asia) decrease in mercury deposition. Moreover, observed long-term temporal trends in mercury wet deposition reflect trends in the global emissions of Hg(0) (Zhang et al., 2016; Weiss-Penzias et al.,

2016; Zhang and Jaeglé, 2013). Our study shows that oxidation of Hg(0) in the upper and middle troposphere is the key to linking the global emissions and deposition of mercury.

We also find that the spatial variation in mercury wet deposition flux at MDN sites is significantly influenced by the variation in the contributions of Hg(II) produced in the upper and middle troposphere. In particular, a large fraction of the upper and middle tropospheric Hg(II) over the U.S. is transported

from the subsiding subtropical anticyclone over the Pacific Ocean. Thus, we expect that variability in the location of the Pacific anticyclone, the synoptic wind patterns transporting Hg(II) to the U.S., the heights of the precipitating clouds, in addition to the amount and type of precipitation can affect Hg wet deposition flux over a particular area. These meteorological conditions vary in response to natural variability associated with multiyear phenomena, such as the El Niño-La Niña cycle (Gratz et al., 2009),

and can confound the interpretation of spatial and temporal trends in wet deposition at MDN sites.

Our results support the idea of a global pool of Hg(II) in the free troposphere. We find that this global pool of Hg(II) is concentrated in the upper troposphere (above 7 km) and extends to lower altitudes in the subsidence areas of the subtropical anticyclones. These regions of the atmosphere are where most of the production of Hg(II) takes place, and where the lifetime of Hg(II) against reduction and deposition

is the longest, making them ideal target regions for future aircraft-based campaigns to understand the chemistry of mercury in the atmosphere.





## 7 Conclusions

We have added to the GEOS-Chem mercury model a Hg(II) tagging method following regions where Hg(II) is produced. We have performed a two-year simulation (2013–2014) with the tagged Hg(II) tracers, and have found that Hg(II) produced in the upper and middle troposphere constitutes 91% of the

tropospheric mass of Hg(II), 91% of the annual Hg(II) wet deposition flux, and 52% of the annual Hg(II) dry deposition flux. The disproportionately high contribution of the Hg(II) produced in these regions is the result of higher production of Hg(II) in the upper and middle troposphere combined with a longer lifetime of Hg(II) and the large-scale subsidence of Hg(II) in the troposphere. Hg(II) produced in the upper and middle troposphere contributes 63% to surface Hg(II) over the continents, and 74–82%

over western U.S, South America, Africa, and Australia. Over the oceans, however, surface Hg(II) is formed locally in the marine boundary layer because of Br released from sea-salt aerosols. Directly emitted anthropogenic Hg(II) makes up a significant fraction (27–69%) of surface Hg(II) concentrations near source regions in eastern U.S., Europe and South and East Asia. However, the wet deposition flux in these regions is largely (~90%) the result of Hg(II) produced in the upper and middle troposphere.

We examined the sensitivity of our results by performing additional simulations with lower Br concentrations, different oxidants ($O_3$ and OH), and different Hg(0):Hg(II) anthropogenic emission speciation. In these simulations, too, we found that Hg(II) produced in the upper and middle troposphere together contribute significantly to the tropospheric Hg(II) mass (78–90%) and the global Hg(II) surface deposition flux (57–76%).

We quantified the role of Hg(II) in dry subtropical anticyclones and found it exerts a strong influence on Hg(II) concentrations at the surface and 500 hPa between 40°S and 40°N. About >60% of the surface Hg(II) over dry areas, such as the western U.S., is transported from these subtropical regions, while 74% of Hg(II) at 500 hPa over the continental U.S. originated in the subtropical anticyclones. We also found that 75% of the observations with Hg(II) concentrations greater than 250 pg m$^{-3}$ observed

during the NOMADSS aircraft campaign were transported from the subsidence regions, compared to only 10% for samples with Hg(II) concentrations less than 100 pg m$^{-3}$.

We examined the consistency of our modeling results with measurements at the MDN, EMEP, and AMNet sites. We found reasonable agreement between the modeled and observed Hg wet deposition





flux at the MDN sites (NMB: -7 to +20%, FAC2: 66 to 88%), and surface Hg(II) concentration at AMNet sites in the eastern U.S. (NMB: -21%, FAC2: 70%), but poorer agreement for Hg wet deposition flux at EMEP observations (NMB: -46%, FAC2: 55%). We also found that the Hg wet deposition flux at the MDN sites increases with increase in precipitation and the contribution of Hg(II)

produced in the upper and middle troposphere. Together, they explain 55% of the spatial variation in the wet deposition flux across the MDN network. For AMNet sites in the eastern U.S., we find that 27% of the spatial variation is explained by the contribution of emitted Hg(II) to surface Hg(II) concentrations. Our results highlight the importance of the upper and middle troposphere as primary sites of Hg(II) production and of the subtropical anticyclones as the primary conduits for the production and export of

Hg(II) in the global atmosphere.

## Acknowledgements

This work was supported by funding from the National Science Foundation under award number 1217010. We thank Dan Jaffe for helpful feedback on the manuscript. We thank Dan Jaffe, Jesse Ambrose and Lynne E. Gratz for the NOMADSS measurements, the National Atmospheric Deposition

Program for the MDN and AMNet measurements, and the European Monitoring and Evaluation Programme for the wet deposition measurements over Europe. We appreciate support from the GEOS-Chem user community.

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



**Table 1 Tropospheric budgets of Hg(II) and individual tagged Hg(II) tracers.**

|  | Total Hg(II) | Tagged Hg(II) tracers[a] | | | | |
|---|---|---|---|---|---|---|
|  |  | UT | MT | LT | STRAT | E-Hg(II) |
| Tropospheric mass of Hg(II)[b] [Mg] | 616 | 514 | 47 | 4 | 50 | 1 |
| Mass located in UT [Mg] | 479 | 429 | 3 | 0 | 47 | 0 |
| Mass located in MT [Mg] | 117 | 78 | 36 | 0 | 3 | 0 |
| Mass located in LT [Mg] | 20 | 7 | 8 | 4 | 0 | 1 |
| Hg(II) production [b] [Mg a$^{-1}$] | 15,790 | 8,560 | 4,190 | 2,460 | 410 | 170 |
| Hg(II) reduction [Mg a$^{-1}$] | 9,740 | 5,750 | 2,390 | 1,260 | 290 | 50 |
| Hg(II) wet deposition [Mg a$^{-1}$] | 3,740 | 2,250 | 1,150 | 230 | 80 | 30 |
| Hg(II) dry deposition [Mg a$^{-1}$] | 2,310 | 570 | 640 | 970 | 40 | 90 |
| Hg(II) lifetime [days] | 14 | 22 | 4.1 | 0.6 | 45 | 2.2 |

(a) Regions are defined as follows: UT (upper troposphere: 400hPa–tropopause), MT (middle troposphere: 750–400hPa), LT (lower troposphere: surface–750hPa), STRAT (stratosphere), E-Hg(II) (directly emitted anthropogenic Hg(II)).

5    (b) 1 Mg = 10$^6$ g, and 1 Mg a$^{-1}$ = 10$^6$ g per year.





**Figure 1: (a) Annual Hg(II) wet deposition fluxes and (b) volume-weighted mean (VWM) mercury concentrations over the U.S. for 2013–2014. The map backgrounds show the GEOS-Chem results and the filled circles show the Mercury Deposition Network (MDN) measurements. The bottom two rows (c and d) show the seasonal variations in wet deposition and VWM concentrations for the four regions marked by white boxes in (a) and (b): west (WE), central (CE), northeast (NE), and southeast (SE). Black circles and error bars show the observed means and standard deviations. The red lines and orange shading are for the modeled means and standard deviations. Each panel displays the Normalized Mean Bias (NMB) and the fraction of model-observation pairs within a factor of 2 of each other (FAC2). The number of stations in each region (N_STA) is also shown. Note the different scales on the y-axis for the WE region relative to the other regions (panels c and d).**

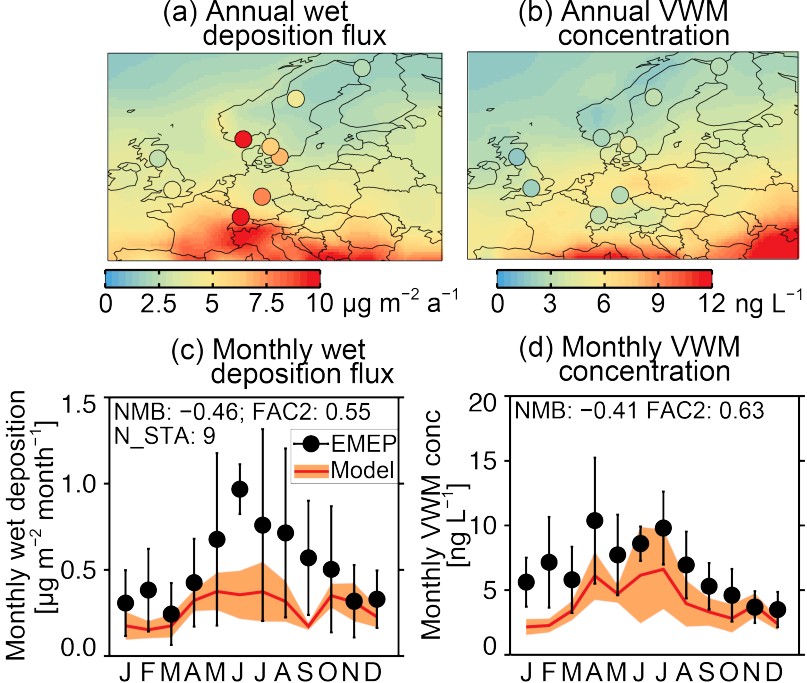

**Figure 2: Same as Figure 1, but for European Monitoring and Evaluation Programme (EMEP) sites.**

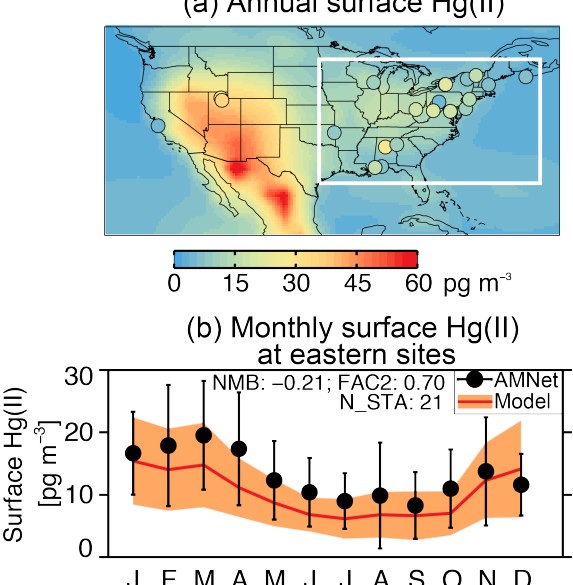

Figure 3: (a) Annual surface Hg(II) concentrations over the U.S. The map backgrounds show the GEOS-Chem concentrations (2013–2014), and the filled circles show the observations at Atmospheric Mercury Network (AMNet) sites (2009–2012). (b) Monthly surface Hg(II) concentrations at the AMNet sites in the eastern U.S. (white box in panel a). Black circles and error bars show the mean and standard deviation of the monthly-mean observations. Red lines and orange shading indicate the modelled means and standard deviations.





**Figure 4: Modeled zonal mean (a) Hg(II) concentrations (pg m⁻³), (b) Hg(II) production rates (pg m⁻³ month⁻¹), and (c) lifetime (days) for 2013-2014. Panels (d-h) show the percent contributions of Hg(II) tagged tracers produced in the upper troposphere (UT), middle troposphere (MT), lower troposphere (LT), stratosphere (STRAT), and directly emitted (E-Hg(II)). Dotted lines indicate our boundaries for STRAT, UT, MT, and LT.**

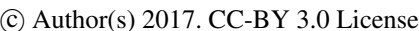



**Figure 5: (a) Annual-mean surface Hg(II) concentration, (b) wet deposition flux, (c) Hg(II) dry deposition flux, and (d) total (wet+dry) deposition flux simulated for 2013-2014. Contributions from tagged Hg(II) tracers to (e) surface Hg(II) concentrations, (f) wet deposition flux, (g) dry deposition flux, and (h) total deposition flux.**





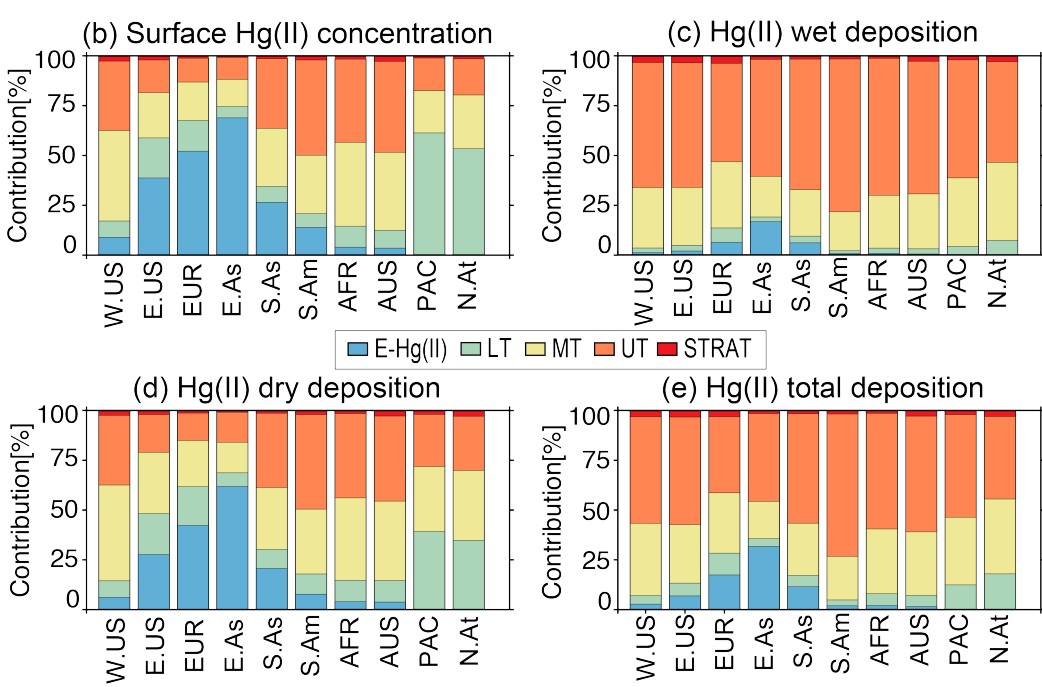

Figure 6: (a) Boundaries and names for regions used in panels b-e. Regional contributions of tagged Hg(II) tracers to (b) Hg(II) surface concentrations, (c) Hg(II) wet deposition, (d) Hg(II) dry deposition, (e) Hg(II) total (wet+dry) deposition. For continental regions the averages are calculated over land only.





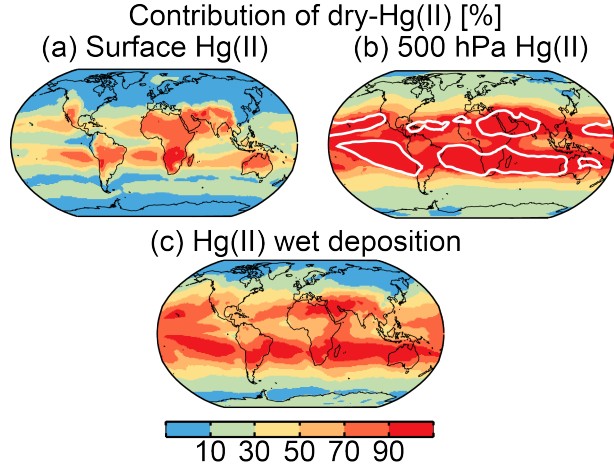

**Figure 7: Annual-mean contribution of dry-Hg(II) to (a) surface Hg(II) concentrations, (b) 500 hPa Hg(II) concentrations, and (c) Hg(II) wet deposition flux for 2013. The white contours in (b) show the boundaries at 500 hPa for areas with RH less than 20% for a minimum of four months of the year.**





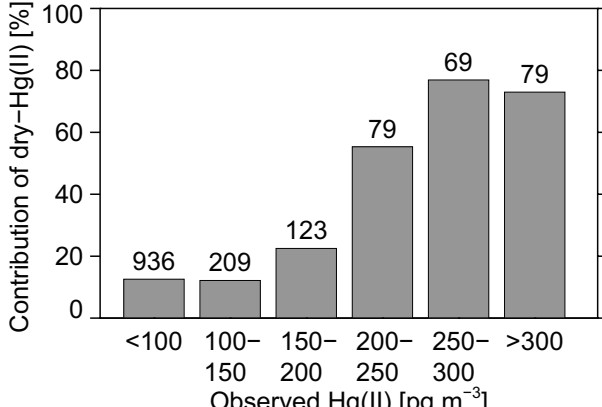

**Figure 8: Modeled contribution of the dry-Hg(II) tracer to observed Hg(II) concentrations during the NOMADSS aircraft campaign. The number of 2.5-minute observations points in each concentration bin is shown on top of the bars.**




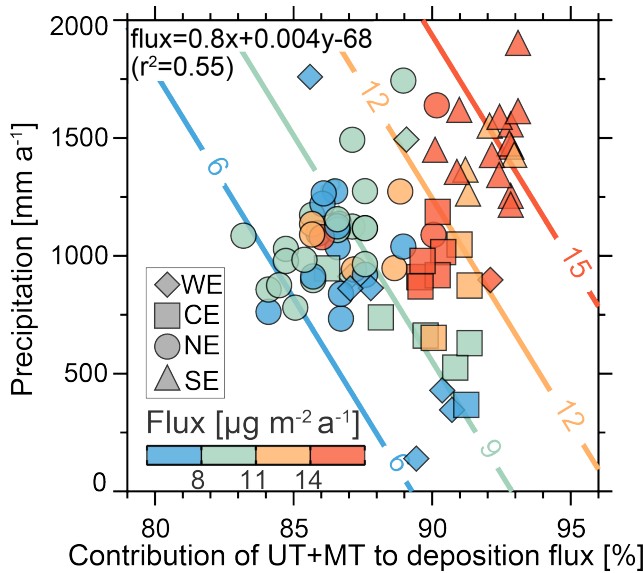

**Figure 9: Relationship of observed MDN Hg wet deposition flux (in units of μg m$^{-2}$ a$^{-1}$) to observed precipitation (mm a$^{-1}$) and modeled contribution of UT and MT tracers to the Hg(II) wet deposition flux (%). The symbols identify MDN sites for each region in Fig 1 (WE: diamonds, CE: squares, NE: circles, and SE: triangles), with color-coding according to observed wet deposition flux. Also shown is the multiple linear regression equation relating flux to the contribution of UT+MT tracers (x) and the observed precipitation (y), and the square of the correlation coefficient (r$^2$). Colored contours correspond to deposition fluxes calculated with the regression equation.**





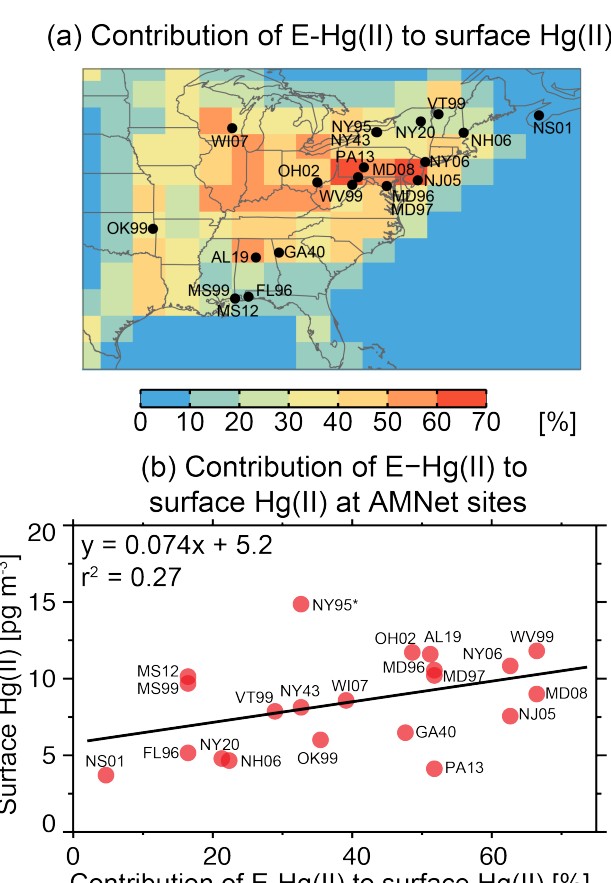

**Figure 10 (a) Simulated surface concentration of Hg(II) for 2013–2014. Also shown are the locations of the AMNet stations mapped to the model grid. (b) Relationship between the 2009–2012 median Hg(II) concentrations observed at the AMNet sites and the contribution of E-Hg(II) tracer to surface Hg(II) concentrations. The black line is the best-fit line from ordinary least squares regression. The text displays the regression equation and the square of the correlation coefficient ($r^2$). The outlier NY95 is excluded from the regression calculation.**