# Peer review of "Subtropical subsidence and surface deposition of oxidized mercury produced in the free troposphere"

_Atmospheric Chemistry and Physics, 2017_

## Referee Comment (RC1) · Anonymous Referee #3 · 12 Mar 2017

Reviewer comments on Shah & Jaegle 2017, "Surface deposition of oxidized mercury dominated by production in the upper and middle troposphere"

12 March 2017

**General comments**
The authors present a modeling study on important regions of Hg(II) production in the troposphere and how those regions contribute to surface deposition. The paper is logically organized and well written. The authors have clearly put a lot of time and thought into the analysis and writing the paper. This will make a valuable contribution to the literature. I recommend minor revisions before publication.

A few general comments to consider:

Consider placing less emphasis on findings about the middle and upper troposphere being important regions for Hg(II) production and deposition, and putting more emphasis on the importance of subtropical anticyclones. It's been established for a while that the free trop is a key region for Hg (II) (flight obs: Franz Slemr, Dan Jaffe, Seth Lyman, Murphy et al. 2006, Brooks et al. 2014; models: Selin & Jacob 2008, Holmes et al. 2010, Bieser et al. 2014, Shah et al., 2016, Horowitz et al., 2016). I would go as far as to consider changing the title of the manuscript to something about subtropical anticyclones -- that's the new, exciting piece and would draw in more readers.

The model spin-up (6 years) is less than half that of other GEOS-Chem Hg model studies (15 years; Holmes et al. 2010 and Horowitz et al. 2016). The rationale for the 15-yr spin-up provided by Holmes and Horowitz is that that's how long it takes to equilibrate the stratosphere. What's the justification for a 6-yr spin up? What are the implications if your model stratosphere hasn't reached equilibrium with the upper troposphere?

Section 3 could be improved by adding more insight and narrative. It presently feels a bit like a core dump of numbers. Having a lot of numbers can be useful, but perhaps might be better served in a table.

Section 6 *Implications* could be merged with Section 7 *Conclusions*. Combining the two sections would help trim some of the redundancy.

**Line-by-line comments**

**Page 1**
Line 18: How is "surface" defined? Is that the first level of the model? Or is it used synonymously with lower troposphere is this context?

Line 25: What accounts for the other 45%? That's surprising precip + Hg(II) production only account for 55%.

Lines 27-28: Statement is unclear. Is there a word missing? "Our simulation points to a large role of Hg(II) present in the dry subtropical subsidence regions…" Confused about the role of Hg(II).

Line 31: "Contribution of these dry regions…" Unclear what the dry regions are contributing to. Hg(II) concentrations? Hg(II) mass in the free troposphere?

Lines 32-34: "Our results highlight the importance of the upper and middle troposphere as key regions for Hg(II) production and of the subtropical anticyclones as the primary conduits for the production and export of Hg(II) to the global atmosphere." I might delete or reword the underlined part. The subtropical anticyclone part is new. I'd play that up in the abstract.

**Page 2**
Line 4: Recommend amending the sentence to say "most aquatic ecosystems".

Line 9: "Global dry deposition fluxes of gaseous elemental mercury (Hg(0)) and oxidized mercury in the gas and particle phases (Hg(II)) are comparable." Needs a citation. Jeroen Sonke's group published work in 2015 or 2016 looking at dry dep in peat. How does your statement line up with the Sonke lab's peat findings?

Line 16: Sproveiri et al. 2010 is a relevant citation.

Line 30: Please quantify "clean" and "dry".
**Page 5**
Lines 3-4: "We assume that stack emissions (emission height > 50m) of Hg consist of 90% Hg(0) and 10% Hg(II)." Needs some justification. Even better if you can include a citation.

**Page 6**
Line 27: Are the assumptions about Hg wet scavenging on lines 15-20 relevant? "Below clouds, gas-phase Hg(II) is washed out by dissolving in falling raindrops (T > 268K), but not in falling snow and ice (Amos et al., 2012). Particle-phase Hg(II) is washed out in collisions in falling rain, snow and ice with different efficiencies (Wang et al., 2011)."

**Page 7**
Lines 10-11: "We adjust the reduction rate to best match aircraft- and ground-based observations of Hg(0) over the mid-latitudes." What rate did you come up with? How does that compare to previous GEOS-Chem modeling studies?

Line 28-29: "...model spin-up period of six years." Is 6 years long enough to spin up the stratosphere? Holmes et al. (2010) and Horowitz et al. (2016) had to initialize their GEOS-Chem simulations with a 15-yr spin-up to equilibrate the stratosphere.

**Page 8**
Line 3: How does the subtropical subsidence in 2013 compare to other years? Was this a dry year with lots of subsidence? Or an average year? A sense of the interannual variability would be helpful.

**Page 11**
Line 28: "...while the contribution from E-Hg(II) is noticeable mainly in East Asia." Please quantify "noticeable".

**Page 13**
Line 13: Please quantify "strong influence".

Line 18: Please quantify "small".

Lines 20-23: How much confidence can be placed in the statement, "Surface Hg(II) in areas poleward of 40° is from anthropogenic emissions (Europe), is produced locally (polar regions)..." give that you have a step function in Br-concentrations at 45 N (Figure 4)?

---

## Referee Comment (RC2) · Anonymous Referee #1 · 15 Mar 2017

General comments

This paper investigates – using the GEOS-Chem global chemical transport model - how surface deposition of divalent mercury species (Hg(II)) is influenced by Hg(II) production at different atmospheric heights. The authors show that surface deposition is dominated by production in the upper and middle troposphere and highlight the large role of subtropical anticyclones as a global reservoir of Hg(II). This study also shows that regional decreases in anthropogenic mercury emissions will not lead to a proportional regional decrease in wet deposition. The paper is organized clearly, easy to follow, well written, and will make a valuable contribution to the literature. However, I find the evaluation of the model with observations insufficient and not up to date. This

paper will be suitable for publication after the authors address the following issues.

Major comments: Comparison with observations

The two-year simulation (2012-2014) is evaluated with ground-based observations of Hg(II) concentrations and wet deposition. Section 2.2.3 concludes that the simulation reproduces quite well the spatial distribution and seasonal cycle of Hg(II) and wet deposition over the US but displays a 46% underestimate of wet deposition observed at EMEP sites. So what? How might this uncertainty affect the distribution of the tagged Hg(II) and ultimately their contributions to wet/dry distribution fluxes in different regions of the world? Additionally, the model is evaluated over the US and Europe only, using ground-based observations. The authors should consider using recent data from ground-based sites, aircraft campaigns and high-altitude sites to evaluate the model in different regions of the world and at different heights. To me, evaluating a model used to investigate the global distribution of Hg(II) at different heights a) over the US only, and b) at ground level only is not convincing enough.

1. Ground-based observations

1.1 Hg(II) concentrations The authors use the 2009-2012 AMNet observations to evaluate the model over the US. I understand that the authors use data that are publicly available. However, evaluating 2013-2014 model outputs with 2009-2012 observations is not satisfying unless inter-annual variability is discussed at some point. In Europe, the authors highlight a discrepancy between modeled/observed wet deposition and suggest that this could "indicate an underestimate in the modeled Hg(II) concentrations over the region". The authors could easily check that since Hg(II) data are available for 2013-2014 (Sprovieri et al., 2016) at Iskrba (Slovenia), Longobucco (Italy), and Rao (Sweden – see also Wängberg et al., 2016). Additionally, how well can the model reproduce Hg(II) concentrations elsewhere? Still according to Sprovieri et al. (2016), Hg(II) data are available around the world for years 2013-2014 at Amsterdam Island (see also Angot et al., 2014), Bariloche (Argentina), Cape Hedo (Japan), Manaus (Brazil), and

Minamata (Japan).

1.2 Wet deposition Same as above, why don't the authors use recent wet deposition data collected around the world to evaluate the model in different regions of the world? A recent paper (Sprovieri et al., 2017) present seasonal and annual variations of Hg wet deposition and concentration collected at 17 ground-based sites in the Northern and Southern Hemispheres as part of the GMOS project. Additionally, page 9, lines 2-4: "Over the southeast US, the modeled VWM concentrations are higher than observations during winter and spring, suggesting a model overestimate in atmospheric Hg(II) concentrations in that region or an overestimate in the amount of Hg(II) scavenged by precipitation". If the model overestimates the amount of Hg(II) scavenged by precipitation, what is the possible influence on results presented in section 3.2, i.e. on the modeled contribution of MT and UT? I would like to see a discussion on how results presented in section 2.2.3 (comparison of modeled and measured Hg(II)) affect results presented thereafter.

2. Vertical profiles

The authors should consider using recent data from aircraft campaigns and high-elevation sites to evaluate the model in different regions of the world. How well can the model reproduce these observations (see for instance Bieser et al., 2016).

2.1 Aircraft campaigns An evaluation of the model is done, over the US, in a previous paper (Shah et al., 2016) during the NOMADSS campaign. The authors could refer to this paper here. Within the GMOS project, vertical profiles were taken on board research aircraft in August 2013 in background air over different locations in Slovenia and Germany (Weigelt et al., 2016). Additionally, Hg(0), Hg(II), and Hg(p) profiles were collected on 28 flights between August 2012 and July 2013 (1000 to 6000 m, Brooks et al., 2014). Finally, the authors could use data from the intercontinental flights between Germany and North/South America under the umbrella of the CARABIC project (Slemr et al., 2014, 2016).

2.2 High-elevation ground sites The authors could use data collected at various high-elevation sites such as Mt. Walinguan (China), Mt. Ailao (China), Kodaicanal (India), Everest/K2 (Nepal) and Col Margherita (Italy) (Sprovieri et al., 2016) to evaluate Hg(0) and/or Hg(II) concentrations. Note that mercury data discussed in this paper are available upon request at: http://sdi.iia.cnr.it/geoint/publicpage/GMOS/gmos.historical.zul.

Other comments: Model sensitivity to oxidation chemistry and emission speciation

The authors perform an additional one-year sensitivity simulation using the original GEOS-Chem Br concentrations instead of the 3 times Br concentrations in the base simulation. Given that updates by Schmidt et al. (2016) have resulted in an improved agreement with satellite and in situ observation of BrO, I wonder why the authors did not perform an additional simulation using these updated fields. Page 9, line 17: "suggesting that the modeled oxidation rate is too slow over this region". Using Br fields from Schmidt et al. (2016), i.e., a factor 2.3 increase in free tropospheric Br concentrations north of 45N might lead to a better agreement between modeled/observed data over Europe.

Page 12, lines 24-33. How do these results compare to the results by Bieser et al. (2016)? According to the latter, "high RM concentrations in the UT could be reproduced by oxidation by Br while elevated concentrations in the LT were better reproduced by OH and ozone". Does it sound feasible and adequate to implement two different mechanisms in GEOS-Chem depending on the altitude?

Line by line comments

Section 2.2: Which version of GEOS-Chem do you use?

Page 8, lines 18-20: "The model reproduces the observed seasonal variations in the central and northeast regions, but underestimates the summer deposition fluxes in the southeast because of a factor of 2 underestimate in summertime precipitation by the GEOS-FP meteorological fields". Is that also the case for other (GEOS-5, MERRA)

meteorological fields? If not, why don't the authors use them? MERRA meteorological data are available for 2013-2014.

Page 9, line 2: there is a typo "Over the southeast US, tmodeled (. . .)".

Page 9, lines 10-12: "(. . .) likely because the upward scaling of the Br concentrations in our simulation did not extend north of 45N and covered only parts of Southern Europe". Could you please add the latitude on the various figures?

Figure 4e: I am just curious; how can you explain the elevated contribution of MT tracer over the Antarctic continent?

Figure 10b: Why is NY95 excluded from the regression calculation? I agree that it is an outlier here, but the question is why? According to info found on AMNet website (and not in the paper. . .), the collection of Hg(II) concentrations stopped in November 2009 at this site. This suggests that the authors only have a few months of data at this site, and not data for the entire 2009-2012 period. That kind of information would be useful (in supplementary?) in order to get a better insight on which observation data are used to evaluate the model.

References

Angot, H., Barret, M., Magand, O., Ramonet, M., Dommergue, A., 2014. A 2-year record of atmospheric mercury species at a background Southern Hemisphere station on Amsterdam Island. Atmos Chem Phys 14, 11461–11473. doi:10.5194/acp-14-11461-2014

Bieser, J., Slemr, F., Ambrose, J., Brenninkmeijer, C., Brooks, S., Dastoor, A., DeSimone, F., Ebinghaus, R., Gencarelli, C., Geyer, B., Gratz, L.E., Hedgecock, I.M., Jaffe, D., Kelley, P., Lin, C.-J., Matthias, V., Ryjkov, A., Selin, N., Song, S., Travnikov, O., Weigelt, A., Luke, W., Ren, X., Zahn, A., Yang, X., Zhu, Y., Pirrone, N., 2016. Multi-model study of mercury dispersion in the atmosphere: Vertical distribution of mercury species. Atmos Chem Phys Discuss 2016, 1–54. doi:10.5194/acp-2016-1074

Brooks, S., Ren, X., Cohen, M., Luke, W.T., Kelley, P., Artz, R., Hynes, A., Landing, W., Martos, B., 2014. Airborne Vertical Profiling of Mercury Speciation near Tullahoma, TN, USA. Atmosphere 5, 557–574. doi:10.3390/atmos5030557

Schmidt, J.A., Jacob, D.J., Horowitz, H.M., Hu, L., Sherwen, T., Evans, M.J., Liang, Q., Suleiman, R.M., Oram, D.E., Le Breton, M., Percival, C.J., Wang, S., Dix, B., Volkamer, R., 2016. Modeling the observed tropospheric BrO background: Importance of multiphase chemistry and implications for ozone, OH, and mercury. J. Geophys. Res. Atmospheres 121, 2015JD024229. doi:10.1002/2015JD024229

Shah, V., Jaeglé, L., Gratz, L.E., Ambrose, J.L., Jaffe, D.A., Selin, N.E., Song, S., Campos, T.L., Flocke, F.M., Reeves, M., Stechman, D., Stell, M., Festa, J., Stutz, J., Weinheimer, A.J., Knapp, D.J., Montzka, D.D., Tyndall, G.S., Apel, E.C., Hornbrook, R.S., Hills, A.J., Riemer, D.D., Blake, N.J., Cantrell, C.A., MauldinÂăIII, R.L., 2016. Origin of oxidized mercury in the summertime free troposphere over the southeastern US. Atmos Chem Phys 16, 1511–1530. doi:10.5194/acp-16-1511-2016

Slemr, F., Weigelt, A., Ebinghaus, R., Brenninkmeijer, C., Baker, A., Schuck, T., Rauthe-Schöch, A., Riede, H., Leedham, E., Hermann, M., van Velthoven, P., Oram, D., O'Sullivan, D., Dyroff, C., Zahn, A., Ziereis, H., 2014. Mercury Plumes in the Global Upper Troposphere Observed during Flights with the CARIBIC Observatory from May 2005 until June 2013. Atmosphere 5, 342–369. doi:10.3390/atmos5020342

Slemr, F., Weigelt, A., Ebinghaus, R., Kock, H.H., Bödewadt, J., Brenninkmeijer, C.A.M., Rauthe-Schöch, A., Weber, S., Hermann, M., Becker, J., Zahn, A., Martinsson, B., 2016. Atmospheric mercury measurements onboard the CARIBIC passenger aircraft. Atmos Meas Tech 9, 2291–2302. doi:10.5194/amt-9-2291-2016

Sprovieri, F., Pirrone, N., Bencardino, M., D'Amore, F., Angot, H., Barbante, C., Brunke, E.-G., Arcega-Cabrera, F., Cairns, W., Comero, S., Diéguez, M.D.C., Dommergue, A., Ebinghaus, R., Feng, X.B., Fu, X., Garcia, P.E., Gawlik, B.M., Hageström, U., Hansson, K., Horvat, M., Kotnik, J., Labuschagne, C., Magand, O., Martin, L., Mashyanov,

N., Mkololo, T., Munthe, J., Obolkin, V., Ramirez Islas, M., Sena, F., Somerset, V., Spandow, P., Vardè, M., Walters, C., Wängberg, I., Weigelt, A., Yang, X., Zhang, H., 2017. Five-year records of mercury wet deposition flux at GMOS sites in the Northern and Southern hemispheres. Atmos Chem Phys 17, 2689–2708. doi:10.5194/acp-17-2689-2017

Sprovieri, F., Pirrone, N., Bencardino, M., D'Amore, F., Carbone, F., Cinnirella, S., Mannarino, V., Landis, M., Ebinghaus, R., Weigelt, A., Brunke, E.-G., Labuschagne, C., Martin, L., Munthe, J., Wängberg, I., Artaxo, P., Morais, F., Barbosa, H.D.M.J., Brito, J., Cairns, W., Barbante, C., Diéguez, M.D.C., Garcia, P.E., Dommergue, A., Angot, H., Magand, O., Skov, H., Horvat, M., Kotnik, J., Read, K.A., Neves, L.M., Gawlik, B.M., Sena, F., Mashyanov, N., Obolkin, V., Wip, D., Feng, X.B., Zhang, H., Fu, X., Ramachandran, R., Cossa, D., Knoery, J., Maruszczak, N., Nerentorp, M., Norstrom, C., 2016. Atmospheric mercury concentrations observed at ground-based monitoring sites globally distributed in the framework of the GMOS network. Atmos Chem Phys 16, 11915–11935. doi:10.5194/acp-16-11915-2016

Wängberg, I., Nerentorp Mastromonaco, M.G., Munthe, J., Gårdfeldt, K., 2016. Airborne mercury species at the Råö background monitoring site in Sweden: distribution of mercury as an effect of long-range transport. Atmos Chem Phys 16, 13379–13387. doi:10.5194/acp-16-13379-2016

Weigelt, A., Ebinghaus, R., Pirrone, N., Bieser, J., Bödewadt, J., Esposito, G., Slemr, F., van Velthoven, P.F.J., Zahn, A., Ziereis, H., 2016. Tropospheric mercury vertical profiles between 500 and 10 000 m in central Europe. Atmos Chem Phys 16, 4135–4146. doi:10.5194/acp-16-4135-2016

---

## Referee Comment (RC3) · Anonymous Referee #2 · 21 Mar 2017

Review of "Surface deposition of oxidized mercury dominated by production in the upper and middle troposphere" by Shah and Jaegle.

The manuscript provides a thorough diagnosis of Hg chemical processing in the lower, middle, and upper troposphere within the GEOS-Chem model. The results are an incremental advance over past work in establishing the important role of oxidation in the middle and upper troposphere. The main advance over past work is in diagnosing the subtropical anticyclones as conduits for supplying Hg(II) to the lower troposphere.

I concur with the other reviewer who commented on the lack of model comparisons to aircraft data and observations in the subtropics, since that is where much of the action in the model is happening. The authors' earlier work with aircraft could be in-

cluded in discussion (Shah et al., 2016). It would also be very helpful to compare the model to surface observations in the subtropics, where they are available (e.g. Sheu et al., 2010). Likewise, I agree with the reviewer who pointed out that simulations for 2013-2014 are compared with AMNet observations for 2009-2012 without discussion of interannual variability.

The title is too sweeping. It implies that Hg(II) emissions and Hg(II) produced in the lower troposphere are minor sources of Hg(II) deposition. While that may be true on a global average basis (Table 1), Figure 5 shows that Hg(II) emissions contribute more than 50% of deposition in major industrial regions and lower troposphere Hg(II) dominates in polar regions. The 2x2.5 degree resolution of the model also likely dilutes the importance of Hg(II) emissions near large sources. These caveats are critical for policymakers, but are not reflected in the title or mentioned in the abstract.

My remaining comments are minor.

Please specify the version of the GEOS-Chem model used in this work.

Eq. R1 has a typo "15" in it.

The rate coefficient k_1f appears to be missing an exponent. Please check all rate expressions

P11, L1: typo: "tmodeled"

Table 1 and P12 report a 45 day lifetime for STRAT Hg(II). That seems surprisingly short considering that nearly zero reduction should happen in the stratosphere, based on the model assumption that reduction requires liquid water clouds. Based on context, I think the authors mean that the lifetime of Hg(II) produced in the stratosphere is 45 days once it enters the troposphere, but this is not clear.

Section 5 addresses the contribution of upper tropospheric Hg(II) to surface deposition across the US. Other recent papers on this topic are Weiss-Penzias et al., (2015); Shanley et al., (2015); Coburn et al., (2016); Kaulfus et al., (2017).

P18L6. The regression equation is not adequately explained. The units of each variable and coefficient must be provided. Is the regression equation fitted to the observed or modeled Hg fluxes?

REFERENCES

Coburn, S., Dix, B., Edgerton, E., Holmes, C. D., Kinnison, D., Liang, Q., Schure, ter, A., Wang, S. and Volkamer, R.: Mercury oxidation from bromine chemistry in the free troposphere over the southeasternÂăUS, Atmos. Chem. Phys., 16(6), 3743–3760, doi:10.5194/acp-16-3743-2016, 2016.

Kaulfus, A. S., Nair, U., Holmes, C. D. and Landing, W. M.: Mercury Wet Scavenging and Deposition Differences by Precipitation Type, Environ. Sci. Technol., 51(5), 2628–2634, doi:10.1021/acs.est.6b04187, 2017.

Sheu, G. R., Lin, N. H., Wang, J. L., Lee, C. T., Yang, C. F. O., and Wang, S. H.: Temporal distribution and potential sources of atmospheric mercury measured at a high-elevation background station in Taiwan, Atmospheric Environment, 44, 2393-2400, 10.1016/j.atmosenv.2010.04.009, 2010.

Shanley, J. B.; Engle, M. a.; Scholl, M.; Krabbenhoft, D. P.; Brunette, R.; Olson, M. L.; Conroy, M. E. High Mercury Wet Deposition at a "clean Air" Site in Puerto Rico. Environ. Sci. Technol. 2015, 49, 12474−12482.

Weiss-Penzias, P., Amos, H. M., Selin, N. E., Gustin, M. S., Jaffe, D. A., Obrist, D., Sheu, G. R. and Giang, A.: Use of a global model to understand speciated atmospheric mercury observations at five high-elevation sites, Atmos. Chem. Phys., 15(3), 1161–1173, doi:10.5194/acp-15-1161-2015, 2015.

---

## Author Comment (AC2) · 18 Jun 2017

Please see our responses in the attached PDF.

Please also note the supplement to this comment:
http://www.atmos-chem-phys-discuss.net/acp-2017-145/acp-2017-145-AC2-supplement.pdf
* * *

---

## Author Comment (AC3) · 18 Jun 2017

Please see our responses in the attached PDF.

Please also note the supplement to this comment:
http://www.atmos-chem-phys-discuss.net/acp-2017-145/acp-2017-145-AC3-supplement.pdf
* * *

---

## Author Response (AR1)

We thank the referee for their thorough review and helpful suggestions. The reviewer's comments (in black) and our responses follow.

**General comments**
This paper investigates – using the GEOS-Chem global chemical transport model - how surface deposition of divalent mercury species (Hg(II)) is influenced by Hg(II) production at different atmospheric heights. The authors show that surface deposition is dominated by production in the upper and middle troposphere and highlight the large role of subtropical anticyclones as a global reservoir of Hg(II). This study also shows that regional decreases in anthropogenic mercury emissions will not lead to a proportional regional decrease in wet deposition. The paper is organized clearly, easy to follow, well written, and will make a valuable contribution to the literature. However, I find the evaluation of the model with observations insufficient and not up to date. This paper will be suitable for publication after the authors address the following issues.

Following the reviewer's recommendation, we have significantly expanded our model-observation comparison to include sites outside the MDN, EMEP and AMNet networks.
*Hg(II) concentrations:* We have added comparisons with 14 sites measuring surface Hg(II) concentrations. These include stations from the worldwide GMOS network and stations in China, Taiwan, Germany, and Canada.
*Hg wet deposition:* We have added comparisons with 14 sites measuring Hg wet deposition. These include stations from the worldwide GMOS network and stations in China, Taiwan and Puerto Rico, US.
*High-elevation sites:* The 14 additional Hg(II) surface sites include 5 high-elevation sites (elevation > 1500m).
*Aircraft-based observations:* We have also added model-measurement comparison for 2 aircraft campaigns: the campaign over Tullahoma, TN and NOMADSS.

These comparisons are shown in Figs. S1, S2 and S3 that will be included in the supplement to the manuscript and are also displayed at the end of this document. A description of these measurements that will be included in the manuscript is below.
"Ground-based measurements of Hg wet deposition and Hg(II) surface concentration have been made as part of the Global Mercury Observations System (GMOS) network (Angot et al., 2014; Wängberg et al., 2016; Sprovieri et al., 2016, 2017; Travnikov et al., 2017), and at sites in Europe (Weigelt et al., 2013), Canada, and East Asia (Sheu et al., 2010; Sheu and Lin, 2013; Fu et al., 2015, 2016). We use the 2013-2014 measurements wherever available, but use all sites with one year or more of observations. We exclude sites in China classified as urban, because of proximity to large Hg(II) sources. We include 14 sites with annual-mean measurements of Hg wet deposition (Table S1), and 14 sites with annual-mean measurements of surface Hg(II) (Table S2)."
"We also include aircraft-based measurements of Hg(II) carried out near Tullahoma, Tennessee, USA from August 2012 to June 2013 (Brooks et al., 2014)."

**Major comments: Comparison with observations**

The two-year simulation (2012-2014) is evaluated with ground-based observations of Hg(II) concentrations and wet deposition. Section 2.2.3 concludes that the simulation reproduces quite well the spatial distribution and seasonal cycle of Hg(II) and wet de- position over the US but displays a 46% underestimate of wet deposition observed at EMEP sites. So what? How might this uncertainty affect the distribution of the tagged Hg(II) and ultimately their contributions to wet/dry distribution fluxes in different regions of the world?

It suggests that the production of Hg(II) in the free-troposphere over Europe is underestimated which would lead to an underestimate in the contribution of UT and MT tracers.
We have added the following paragraph to Sect. 3.2 in response to this and your other similar question below.
"In Sect. 2.2.3 we saw that the model overestimated observed wet deposition of Hg(II) over southeast U.S. during winter and spring. As a result, our estimate of the contribution of UT and MT tracers is likely an overestimate for this region and season. From our model evaluation, we had also concluded that our free-tropospheric Hg(II) production was too slow over Europe and, possibly, other regions north of 45°N. This suggests an underestimate of the concentrations of modeled UT and MT tracers in these regions."

Additionally, the model is evaluated over the US and Europe only, using ground-based observations. The authors should consider using recent data from ground-based sites, aircraft campaigns and high-altitude sites to evaluate the model in different regions of the world and at different heights. To me, evaluating a model used to investigate the global distribution of Hg(II) at different heights a) over the US only, and b) at ground level only is not convincing enough.
To address this concern, we have significantly expanded our model-observation comparison to include 14 additional stations measuring Hg wet deposition, 14 additional stations measuring Hg(II) surface concentrations, and 2 aircraft-based campaigns (the campaign over Tullahoma, TN and NOMADSS). Tables S1 and S2, and Figs. S1, S2 and S3 displayed at the end of the document will be included in the supplement to the manuscript.

**1. Ground-based observations**
*1.1 Hg(II) concentrations* The authors use the 2009-2012 AMNet observations to evaluate the model over the US. I understand that the authors use data that are publicly available. However, evaluating 2013-2014 model outputs with 2009-2012 observations is not satisfying unless inter-annual variability is discussed at some point.
Good point. From the 4-year (2013-16) "dry-Hg(II)" simulation, we find that the variation in the modeled 2-year average Hg(II) concentrations at the AMNet sites vary by ± 30%.
We have added the following to Sect 2.3.3 "Comparing observations and simulations for different time periods adds additional uncertainty due to inter-annual variations. From four years of model simulation (2013-16), we estimate this uncertainty at ±30%."

In Europe, the authors highlight a discrepancy between modeled/observed wet deposition and suggest that this could "indicate an underestimate in the modeled Hg(II) concentrations over the region". The authors could easily check that since Hg(II) data are available for 2013-2014 (Sprovieri et al., 2016) at Iskrba (Slovenia), Longobucco (Italy), and Rao (Sweden – see also Wängberg et al., 2016). Additionally, how well can the model reproduce Hg(II) concentrations elsewhere? Still according to Sprovieri et al. (2016), Hg(II) data are available around the world for years 2013-2014 at Amsterdam Island (see also Angot et al., 2014), Bariloche (Argentina), Cape Hedo (Japan), Manaus (Brazil), and Minamata (Japan).
We have expanded our model-observation comparison to include 14 additional stations measuring Hg(II) surface concentration, which include GMOS sites for which Hg(II) observations have been published. See Tables S2 and Fig. S2. These comparisons show a reasonable model performance (NMB:-9%, FAC2:50%). Modeled surface Hg(II) concentrations at Råö and Longobucco are in good agreement with the observations (Fig. S2). However, it should be noted that an underestimate in Hg wet deposition reflects an underestimate in the abundance of Hg(II) in the precipitating column which is 1-5 km high typically, and may not be detected from the surface Hg(II) measurements. Comparison with wet deposition measurements at Iskrba and Mace Head also show a model underestimate of 25% in Hg concentration in wet deposition.

*1.2 Wet deposition* Same as above, why don't the authors use recent wet deposition data collected around the world to evaluate the model in different regions of the world? A recent paper (Sprovieri et al., 2017) present seasonal and annual variations of Hg wet deposition and concentration collected at 17 ground-based sites in the Northern and Southern Hemispheres as part of the GMOS project.

We have significantly expanded our model-observation comparison to include 14 additional stations measuring Hg wet deposition, which include GMOS sites for which Hg wet deposition measurements have been published in Sprovieri et al. (2017). See Tables S1 and Fig. S1. The model reproduces the wet deposition observations with a NMB of 52% and FAC2 of 64%, and the VWM concentrations with a NMB of 48% and FAC2 of 78%.

Additionally, page 9, lines 2-4: "Over the southeast US, the modeled VWM concentrations are higher than observations during winter and spring, suggesting a model overestimate in atmospheric Hg(II) concentrations in that region or an overestimate in the amount of Hg(II) scavenged by precipitation". If the model overestimates the amount of Hg(II) scavenged by precipitation, what is the possible influence on results presented in section 3.2, i.e. on the modeled contribution of MT and UT? I would like to see a discussion on how results presented in section 2.2.3 (comparison of modeled and measured Hg(II)) affect results presented thereafter.

See our response to a similar question above.

**2. Vertical profiles**

The authors should consider using recent data from aircraft campaigns and high- elevation sites to evaluate the model in different regions of the world. How well can the model reproduce these observations (see for instance Bieser et al., 2016).

*2.1 Aircraft campaigns* An evaluation of the model is done, over the US, in a previous paper (Shah et al., 2016) during the NOMADSS campaign. The authors could refer to this paper here. Within the GMOS project, vertical profiles were taken on board research aircraft in August 2013 in background air over different locations in Slovenia and Germany (Weigelt et al., 2016). Additionally, Hg(0), Hg(II), and Hg(p) profiles were collected on 28 flights between August 2012 and July 2013 (1000 to 6000 m, Brooks et al., 2014). Finally, the authors could use data from the intercontinental flights between Germany and North/South America under the umbrella of the CARABIC project (Slemr et al., 2014, 2016).

We have expanded our model-observation comparison to include two aircraft-based campaigns (NOMADSS and the one of over Tullahoma, TN). The model captures the Hg(II) vertical profiles observed during these two aircraft campaigns. See Fig. S3. For observations over Tullahoma, TN we find the model NMB of 14% and FAC2 of 52%. For observations above 4 km in the NOMADSS campaign, the model NMB is -29% and FAC2 is 53%.

*2.2 High-elevation ground sites* The authors could use data collected at various high- elevation sites such as Mt. Walinguan (China), Mt. Ailao (China), Kodaicanal (India), Everest/K2 (Nepal) and Col Margherita (Italy) (Sprovieri et al., 2016) to evaluate Hg(0) and/or Hg(II) concentrations. Note that mercury data discussed in this paper are available upon request at: http://sdi.iia.cnr.it/geoint/publicpage/GMOS/gmos.historical.zul.

Five of the 14 additional Hg(II) sites are at high-elevations. See Table S2 and Fig. S2. We find that in general the model captures the relatively higher concentrations observed at these high-elevation sites.

Other comments: Model sensitivity to oxidation chemistry and emission speciation
The authors perform an additional one-year sensitivity simulation using the original GEOS-Chem Br concentrations instead of the 3 times Br concentrations in the base simulation. Given that

updates by Schmidt et al. (2016) have resulted in an improved agreement with satellite and in situ observation of BrO, I wonder why the authors did not perform an additional simulation using these updated fields. Page 9, line 17: "suggesting that the modeled oxidation rate is too slow over this region". Using Br fields from Schmidt et al. (2016), i.e., a factor 2.3 increase in free tropospheric Br concentrations north of 45N might lead to a better agreement between modeled/observed data over Europe.

The bromine fields from Schmidt et al. (2016) have just recently been incorporated into the GEOS-Chem Hg simulation (Horowitz et al., 2017). Therefore, we weren't able to use those fields in our simulations.

Page 12, lines 24-33. How do these results compare to the results by Bieser et al. (2016)? According to the latter, "high RM concentrations in the UT could be reproduced by oxidation by Br while elevated concentrations in the LT were better reproduced by OH and ozone". Does it sound feasible and adequate to implement two different mechanisms in GEOS-Chem depending on the altitude?

Bieser et al. (2017) did not investigate the vertical profile with the OH/O3 oxidation mechanism in GEOS-Chem. They also show that the inter-model variation in the simulated Hg(II) concentrations is larger than the observed variation in the Hg(II) vertical profiles, thus not providing much support for considering an altitude-dependent mechanism in GEOS-Chem. The results of our simulation, using Br chemistry, show good agreement with the aircraft-based observations over Tullahoma, TN (Fig. S3 panel a) and with surface observations at AMNet sites (Fig. 3)

**Line by line comments**
Section 2.2: Which version of GEOS-Chem do you use?
It is v9-02. We have added the following sentence to the manuscript in Sect. 2.2: "We use GEOS-Chem v9-02 (http://acmg.seas.harvard.edu/geos/)."

Page 8, lines 18-20: "The model reproduces the observed seasonal variations in the central and northeast regions, but underestimates the summer deposition fluxes in the southeast because of a factor of 2 underestimate in summertime precipitation by the GEOS-FP meteorological fields". Is that also the case for other (GEOS-5, MERRA) meteorological fields? If not, why don't the authors use them? MERRA meteorological data are available for 2013-2014.

The MERRA precipitation over the SE US during summer is closer to observations. Although it is not possible for us to redo the model setup and simulations with a new meteorological field for this study, it is something we can investigate fully in the GEOS-Chem Hg simulation in the future.

Page 9, line 2: there is a typo "Over the southeast US, tmodeled (. . .)".
Fixed the typo.

Page 9, lines 10-12: "(. . .) likely because the upward scaling of the Br concentrations in our simulation did not extend north of 45N and covered only parts of Southern Europe". Could you please add the latitude on the various figures?
We have added latitude and longitude grids to maps in Figures 1, 2, 3, 5, and 7.

Figure 4e: I am just curious; how can you explain the elevated contribution of MT tracer over the Antarctic continent?
The high elevation of the Antarctic (~2500 m) means that much of the surface is higher than the upper boundary of the lower troposphere (defined here as region below 750 hPa), thus we see elevated contribution from the MT tracer.

Figure 10b: Why is NY95 excluded from the regression calculation? I agree that it is an outlier here, but the question is why? According to info found on AMNet website (and not in the paper. . .), the collection of Hg(II) concentrations stopped in November 2009 at this site. This suggests that the authors only have a few months of data at this site, and not data for the entire 2009-2012 period. That kind of information would be useful (in supplementary?) in order to get a better insight on which observation data are used to evaluate the model.

We agree, and have added two tables in the supplement (Tables S1 and S2, also included at the end of the document) with the details of sites, including their measurement time periods, used in the paper.

The NY95 site was operational from 2009 to 2012. We can't tell why it is an outlier. It is possible that we are missing a Hg(II) emission source close to the site. However, we would like to refer to Gay et al. (2013), page 11345, for a discussion on the GOM and PBM variations at the AMNet sites.

We thank the referee for their thorough review and helpful suggestions. The reviewer's comments (in black) and our responses follow.

The manuscript provides a thorough diagnosis of Hg chemical processing in the lower, middle, and upper troposphere within the GEOS-Chem model. The results are an incremental advance over past work in establishing the important role of oxidation in the middle and upper troposphere. The main advance over past work is in diagnosing the subtropical anticyclones as conduits for supplying Hg(II) to the lower troposphere.

I concur with the other reviewer who commented on the lack of model comparisons to aircraft data and observations in the subtropics, since that is where much of the action in the model is happening. The authors' earlier work with aircraft could be included in discussion (Shah et al., 2016). It would also be very helpful to compare the model to surface observations in the subtropics, where they are available (e.g. Sheu et al., 2010).
We have significantly expanded our model-observation comparison to include 14 additional stations measuring Hg wet deposition, 14 additional stations measuring Hg(II) surface concentrations, some of which are in the subtropics, and two-aircraft based campaigns where Hg(II) was measured (the campaign over Tullahoma, TN and NOMADSS). Tables S1 and S2, and Figs. S1, S2 and S3 at the end of the document will be included in the supplement to the manuscript.

Likewise, I agree with the reviewer who pointed out that simulations for 2013-2014 are compared with AMNet observations for 2009-2012 without discussion of interannual variability.
Good point. From the 4-year (2013-16) "dry-Hg(II)" simulation, we find that the variation in the modeled 2-year average Hg(II) concentrations at the AMNet sites vary by ± 30%.
We have added the following to Sect 2.3.3 "Comparing observations and simulations for different time periods adds additional uncertainty due to inter-annual variations. From four years of model simulation (2013-16), we estimate this uncertainty at ±30%."

The title is too sweeping. It implies that Hg(II) emissions and Hg(II) produced in the lower troposphere are minor sources of Hg(II) deposition. While that may be true on a global average basis (Table 1), Figure 5 shows that Hg(II) emissions contribute more than 50% of deposition in major industrial regions and lower troposphere Hg(II) dominates in polar regions. The 2x2.5 degree resolution of the model also likely dilutes the importance of Hg(II) emissions near large sources. These caveats are critical for policymakers, but are not reflected in the title or mentioned in the abstract.
Excellent point. We have revised the title such that the importance of emissions and production in the lower troposphere is not diminished, and the role of the subtropics is highlighted. The new title is: "Subtropical subsidence and surface deposition of oxidized mercury produced in the free troposphere."
We have also clarified the limitations of the coarse resolution of the global model by adding to the abstract the following (underlined text added):
 "…whereas 26–66% of surface Hg(II) over the eastern U.S., Europe, East Asia, and South Asia is directly emitted. The influence of directly emitted Hg(II) near emissions sources is likely higher, but cannot be quantified by our coarse-resolution global model(2° latitude × 2.5° longitude). Over the oceans…"
In Sect. 3.2 we have added the underlined text:
"We calculate that 27–69% of surface Hg(II) in eastern U.S., Europe, East and South Asia consists of E-Hg(II) (Fig. 6b). The contribution of E-Hg(II) is 80% of higher in areas close to

emission sources (Fig. 5e), and can be even higher within tens of kilometers of the sources. However, the near-source contribution of emitted Hg(II) cannot be estimated with our 2° latitude × 2.5° longitude global model. "
And, in the conclusion, we have added the following underlined text:
"…the wet deposition flux in these regions is largely (~90%) the result of Hg(II) produced in the upper and middle troposphere. The contribution of directly emitted Hg(II) can be higher within tens of kilometers of a source, but cannot be quantified by our coarse-resolution global model."

My remaining comments are minor.
Please specify the version of the GEOS-Chem model used in this work.
It is v9-02.
We have made the following change to Sect. 2.2: "…resolution of 2° latitude × 2.5° longitude and 47 vertical levels for the GEOS-Chem simulations in this study. We use GEOS-Chem v9-02 (http://acmg.seas.harvard.edu/geos/). Global anthropogenic emissions…"

Eq. R1 has a typo "15" in it.
Fixed the typo.

The rate coefficient k_1f appears to be missing an exponent. Please check all rate expressions
Fixed the error and double-checked the rate expressions. They now are as follows:

$$k_{1f} = 1.46 \times 10^{-32} \times \left( \frac{T}{298} \right)^{-1.86} \times [M] \quad cm^3 \ molecule^{-1} \ s^{-1}$$

$$k_{1r} = 2.67 \times 10^{41} \times \exp\left( \frac{-7292}{T} \right) \times \left( \frac{T}{298} \right)^{1.76} \times k_{1f} \quad s^{-1}$$

$$k_2 = 3.9 \times 10^{-11} \quad cm^3 \ molecule^{-1} \ s^{-1}$$

$$k_3 = 2.5 \times 10^{-10} \times \left( \frac{T}{298} \right)^{-0.57} \quad cm^3 \ molecule^{-1} \ s^{-1}$$

P11, L1: typo: "tmodeled"
Fixed the typo.

Table 1 and P12 report a 45 day lifetime for STRAT Hg(II). That seems surprisingly short considering that nearly zero reduction should happen in the stratosphere, based on the model assumption that reduction requires liquid water clouds. Based on context, I think the authors mean that the lifetime of Hg(II) produced in the stratosphere is 45 days once it enters the troposphere, but this is not clear.
That is right. The lifetime is for the STRAT Hg(II) present in the troposphere.
On P12 we have clarified this as follows: "As summarized in Table 1, we find that the tropospheric lifetime of Hg(II)…" (underlined text added). We have changed the last row of Table 1 to: "Hg(II) tropospheric lifetime [days]"

Section 5 addresses the contribution of upper tropospheric Hg(II) to surface deposition across the US. Other recent papers on this topic are Weiss-Penzias et al., (2015); Shanley et al., (2015); Coburn et al., (2016); Kaulfus et al., (2017).
We have now added citations to these papers in the manuscript.

P18L6. The regression equation is not adequately explained. The units of each variable and coefficient must be provided. Is the regression equation fitted to the observed or modeled Hg fluxes?
The regression equation is now clarified, with units for the variables and the coefficients. It is fitted to the observed fluxes.

**Response to referee # 3 comments**
We thank the referee for their thorough review and helpful suggestions. The reviewer's comments (in black) and our responses follow.

**General comments**
The authors present a modeling study on important regions of Hg(II) production in the troposphere and how those regions contribute to surface deposition. The paper is logically organized and well written. The authors have clearly put a lot of time and thought into the analysis and writing the paper. This will make a valuable contribution to the literature. I recommend minor revisions before publication.

A few general comments to consider:
Consider placing less emphasis on findings about the middle and upper troposphere being important regions for Hg(II) production and deposition, and putting more emphasis on the importance of subtropical anticyclones. It's been established for a while that the free trop is a key region for Hg (II) (flight obs: Franz Slemr, Dan Jaffe, Seth Lyman, Murphy et al. 2006, Brooks et al. 2014; models: Selin & Jacob 2008, Holmes et al. 2010, Bieser et al. 2014, Shah et al., 2016, Horowitz et al., 2016). I would go as far as to consider changing the title of the manuscript to something about subtropical anticyclones -- that's the new, exciting piece and would draw in more readers.

Excellent suggestion. We have revised the title to: "**Subtropical subsidence and surface deposition of oxidized mercury produced in the free troposphere.**" We have also extended our "dry-Hg(II)" simulation from one year to four (2013-2016), to get a sense of the interannual variability. We have also modified the abstract and the text to place more emphasis on the subtropical anticyclones.

The model spin-up (6 years) is less than half that of other GEOS-Chem Hg model studies (15 years; Holmes et al. 2010 and Horowitz et al. 2016). The rationale for the 15-yr spin-up provided by Holmes and Horowitz is that that's how long it takes to equilibrate the stratosphere. What's the justification for a 6-yr spin up? What are the implications if your model stratosphere hasn't reached equilibrium with the upper troposphere?

In the revised manuscript, we have conducted a 15-year spin up period and have updated the figures and tables accordingly. The revised version of Table 1 is below. We see only a small (5% or lower) change in the tropospheric budgets of the STRAT, UT, and MT tracers.

**Table 1 Tropospheric budgets of Hg(II) and individual tagged Hg(II) tracers.**

| | Total Hg(II) | Tagged Hg(II) tracers[a] | | | | |
| --- | --- | --- | --- | --- | --- | --- |
| | | UT | MT | LT | STRAT | E-Hg(II) |
| Tropospheric mass of Hg(II)[b] [Mg] | 618 | 517 | 48 | 4 | 48 | 1 |
| Mass located in UT [Mg] | 480 | 432 | 3 | 0 | 45 | 0 |
| Mass located in MT [Mg] | 118 | 79 | 36 | 0 | 3 | 0 |
| Mass located in LT [Mg] | 20 | 7 | 8 | 4 | 0 | 1 |
| Hg(II) production [b] [Mg a$^{-1}$] | 15,790 | 8,560 | 4,190 | 2,460 | 410 | 170 |
| Hg(II) reduction [Mg a$^{-1}$] | 9,740 | 5,750 | 2,390 | 1,260 | 290 | 50 |
| Hg(II) wet deposition [Mg a$^{-1}$] | 3,740 | 2,250 | 1,150 | 230 | 80 | 30 |
| Hg(II) dry deposition [Mg a$^{-1}$] | 2,310 | 570 | 640 | 970 | 40 | 90 |
| Hg(II) tropospheric lifetime [days] | 14 | 22 | 4.1 | 0.6 | 43 | 2.2 |

Section 3 could be improved by adding more insight and narrative. It presently feels a bit like a core dump of numbers. Having a lot of numbers can be useful, but perhaps might be better served

in a table.

Good suggestion. We have deleted some of the redundant numbers from Section 3.1 that were already listed in Table 1, and added more explanation of our results. We have also added another table (see below) to summarize the results of the sensitivity simulations and reduce the numbers written in Sect. 3.3.

**Table 2 Contribution of tagged Hg(II) tracers to the tropospheric mass and total deposition of Hg(II) for the base case and the sensitivity simulations.**

| | Simulation | Tagged Hg(II) tracer contribution [%] | | | | |
| --- | --- | --- | --- | --- | --- | --- |
| | | UT | MT | LT | STRAT | E-Hg(II) |
| Contribution to Hg(II) tropospheric mass [%] | Base | 84 | 8 | <1 | 8 | <1 |
| | Lower UT+MT Br[a] | 71 | 7 | 1 | 21 | <1 |
| | O$_3$/OH oxidation[b] | 61 | 18 | 4 | 17 | <1 |
| | Higher Hg(II) emissions[c] | 84 | 8 | <1 | 8 | <1 |
| Contribution to Hg(II) deposition [%] | Base | 47 | 30 | 19 | 2 | 2 |
| | Lower UT+MT Br[a] | 43 | 21 | 27 | 6 | 3 |
| | O$_3$/OH oxidation[b] | 20 | 38 | 38 | 2 | 2 |
| | Higher Hg(II) emissions[c] | 49 | 28 | 17 | 2 | 4 |

(a) Simulation using the original GEOS-Chem Br concentrations instead of the 3 times Br concentrations in the base simulation,
(b) Simulation using O$_3$ and OH as the Hg(0) oxidants instead of Br as in the base simulation,
(c) Simulation using the default UNEP/AMAP Hg(0):Hg(II) emission speciation of 55%:45% instead of the 90%:10% speciation as in the base simulation.

Section 6 *Implications* could be merged with Section 7 *Conclusions*. Combining the two sections would help trim some of the redundancy.

Following your suggestion, we have trimmed Sect. 6 of the most of the redundancy, but have decided to keep the *Implications* separate from the *Conclusions*.

**Line-by-line comments**
**Page 1**
Line 18: How is "surface" defined? Is that the first level of the model? Or is it used synonymously with lower troposphere is this context?

The surface is defined here as the first level in the model.

Line 25: What accounts for the other 45%? That's surprising precip + Hg(II) production only account for 55%.

We don't know. It could be because of variations in the amount of Hg(II) in the precipitating column, caused by spatial variation in production and loss rates. Hg(II) column amounts in the model are only moderately correlated to the *contribution* of the UT+MT tracers. They are almost perfectly correlated, of course, with the total amounts of UT+MT tracers.

Lines 27-28: Statement is unclear. Is there a word missing? "Our simulation points to a large role of Hg(II) present in the dry subtropical subsidence regions..." Confused about the role of Hg(II).

We have modified the text as follows: "Our simulation points to a large role of  the dry subtropical subsidence regions. Hg(II) present in these regions  accounts for…"

Line 31: "Contribution of these dry regions..." Unclear what the dry regions are contributing to. Hg(II) concentrations? Hg(II) mass in the free troposphere?

Modified to: "…the contribution of Hg(II) from the  dry subtropical regions was found…"

Lines 32-34: "Our results highlight the importance of the upper and middle troposphere as key regions for Hg(II) production and of the subtropical anticyclones as the primary conduits for the production and export of Hg(II) to the global atmosphere." I might delete or reword the underlined part. The subtropical anticyclone part is new. I'd play that up in the abstract.

We have deleted the underlined phrase to emphasize the anticyclone part. The modified sentence is "Our results highlight the importance the subtropical anticyclones as the primary conduits for the production and export of Hg(II) to the global atmosphere."

**Page 2**
Line 4: Recommend amending the sentence to say "most aquatic ecosystems".
We have modified the text accordingly.

Line 9: "Global dry deposition fluxes of gaseous elemental mercury (Hg(0)) and oxidized mercury in the gas and particle phases (Hg(II)) are comparable." Needs a citation. Jeroen Sonke's group published work in 2015 or 2016 looking at dry dep in peat. How does your statement line up with the Sonke lab's peat findings?
Our statement is based on model estimates, and we have added relevant citations.

Line 16: Sproveiri et al. 2010 is a relevant citation.
Agreed. We have added the citation.

Line 30: Please quantify "clean" and "dry".
Clean and dry is defined as RH below 35% and CO below 75 ppbv. We have specified this in the text now.

**Page 5**
Lines 3-4: "We assume that stack emissions (emission height > 50m) of Hg consist of 90% Hg(0) and 10% Hg(II)." Needs some justification. Even better if you can include a citation.
We have some discussion about this assumption in Sect 2.2.1, including citations. We have added a reference to Sect. 2.2.1 to the sentence in question here.

**Page 6**
Line 27: Are the assumptions about Hg wet scavenging on lines 15-20 relevant? "Below clouds, gas-phase Hg(II) is washed out by dissolving in falling raindrops (T > 268K), but not in falling snow and ice (Amos et al., 2012). Particle-phase Hg(II) is washed out in collisions in falling rain, snow and ice with different efficiencies (Wang et al., 2011)."
Since Hg(II) wet deposition is an important part of our work, we wanted to state all relevant model assumptions in the manuscript. These assumptions affect the simulated wet deposition flux and the vertical distribution of Hg(II).

**Page 7**
Lines 10-11: "We adjust the reduction rate to best match aircraft- and ground-based observations of Hg(0) over the mid-latitudes." What rate did you come up with? How does that compare to previous GEOS-Chem modeling studies?
The reduction rate is scaled to the photolysis rate of $NO_2$. We use a scaling coefficient of 0.1, which is a 16 times higher than the reduction rate used by Zhang et al. (2012).

Line 28-29: "...model spin-up period of six years." Is 6 years long enough to spin up the stratosphere? Holmes et al. (2010) and Horowitz et al. (2016) had to initialize their GEOS-Chem

simulations with a 15-yr spin-up to equilibrate the stratosphere.
We have now extended the spin-up period to 15 years.

**Page 8**
Line 3: How does the subtropical subsidence in 2013 compare to other years? Was this a dry year with lots of subsidence? Or an average year? A sense of the interannual variability would be helpful.
To provide a sense of the interannual variability, we have extended our "dry-Hg(II)" simulation from one year to four, and have shown the variation (as anomaly) in Fig. 7. The revised Fig. 7 is below:

[Figure]

Figure 7: Mean and anomaly (maximum deviation from the mean) of the contributions of dry-Hg(II) to (a,b) surface Hg(II) concentrations, (c,d) 500 hPa Hg(II) concentrations, and (e,f) Hg(II) wet deposition flux for 2013-2016. The white contours in (c,d) show the boundaries at 500 hPa for areas with 2013-2016 RH less than 20% for a minimum of four months of the year.

**Page 11**
Line 28: "...while the contribution from E-Hg(II) is noticeable mainly in East Asia." Please quantify "noticeable".
We've changed the sentence as follows: "is greater than 10% mainly in over East Asia"

**Page 13**
Line 13: Please quantify "strong influence". Line 18: Please quantify "small".
The contribution of Hg(II) to surface deposition is 45 ± 25%.
We have added the following text to Line 13 quantifying the strong influence. "We see from Fig. 7a that dry-Hg(II) exerts a disproportionate influence on surface Hg(II) concentrations between 40°S and 40°N, where its contribution is 45 ± 25%."
And, on Line 18, by small we mean <20%. This has been clarified in the revised manuscript.

Lines 20-23: How much confidence can be placed in the statement, "Surface Hg(II) in areas poleward of 40° is from anthropogenic emissions (Europe), is produced locally (polar regions)..." give that you have a step function in Br-concentrations at 45 N (Figure 4)?
This higher Br in the subtropics should increase, if anything, the contribution of dry-Hg(II) poleward of 40°. That we don't see much of a contribution of dry-Hg(II), indicates other processes are involved. This statement relies on both Figures 5 and 7, and to clarify this we are citing both figures in the revised manuscript.

**Table S1: List of stations with observations of Hg wet deposition used in this study**

| Site ID | Site Name | Latitude | Longitude | Elevation (m.a.s.l.) | Measurement period | Network/ Region |
|---|---|---|---|---|---|---|
| CO96 | Molas Pass | 37.75 | -107.69 | 3248 | 2013-2014 | MDN[a] |
| FL11 | Everglades National Park-Research Center | 25.39 | -80.68 | 2 | 2013-2014 | MDN |
| WA18 | Seattle/NOAA | 47.68 | -122.26 | 11 | 2013-2014 | MDN |
| TX21 | Longview | 32.38 | -94.71 | 103 | 2013-2014 | MDN |
| VT99 | Underhill | 44.53 | -72.87 | 399 | 2013-2014 | MDN |
| VA28 | Shenandoah National Park-Big Meadows | 38.52 | -78.43 | 1072 | 2013-2014 | MDN |
| WI36 | Trout Lake | 46.05 | -89.65 | 509 | 2013-2014 | MDN |
| WI99 | Lake Geneva | 42.58 | -88.50 | 288 | 2013-2014 | MDN |
| PA29 | Kane Experimental Forest | 41.60 | -78.77 | 618 | 2013-2014 | MDN |
| PA42 | Leading Ridge | 40.66 | -77.94 | 287 | 2013-2014 | MDN |
| PA72 | Milford | 41.33 | -74.82 | 212 | 2013-2014 | MDN |
| TN11 | Great Smoky Mountains | 35.66 | -83.59 | 640 | 2013-2014 | MDN |
| MN18 | Fernberg | 47.95 | -91.50 | 524 | 2013-2014 | MDN |
| ME02 | Bridgton | 44.11 | -70.73 | 222 | 2013-2014 | MDN |
| ME96 | Casco Bay-Wolfe's Neck Farm | 43.83 | -70.06 | 15 | 2013-2014 | MDN |
| NC08 | Waccamaw State Park | 34.26 | -78.48 | 10 | 2013-2014 | MDN |
| PA13 | Allegheny Portage Historic Site | 40.46 | -78.56 | 739 | 2013-2014 | MDN |
| PA90 | Hills Creek State Park | 41.80 | -77.19 | 476 | 2013-2014 | MDN |
| SC19 | Congaree Swamp | 33.81 | -80.78 | 34 | 2013-2014 | MDN |
| IL11 | Bondville | 40.05 | -88.37 | 212 | 2013-2014 | MDN |
| FL34 | Everglades Nutrient Removal Project | 26.66 | -80.40 | 10 | 2013-2014 | MDN |
| FL05 | Chassahowitzka National Wildlife Refuge | 28.75 | -82.56 | 3 | 2013-2014 | MDN |
| GA09 | Okefenokee National Wildlife Refuge | 30.74 | -82.13 | 45 | 2013-2014 | MDN |
| PA00 | Arendtsville | 39.92 | -77.31 | 269 | 2013-2014 | MDN |
| KS32 | Lake Scott State Park | 38.67 | -100.92 | 863 | 2013-2014 | MDN |
| ME98 | Acadia National Park-McFarland Hill | 44.38 | -68.26 | 150 | 2013-2014 | MDN |
| ME00 | Caribou | 46.87 | -68.01 | 191 | 2013-2014 | MDN |
| ME09 | Greenville Station | 45.49 | -69.66 | 322 | 2013-2014 | MDN |
| MN16 | Marcell Experimental Forest | 47.53 | -93.47 | 431 | 2013-2014 | MDN |
| MN23 | Camp Ripley | 46.25 | -94.50 | 410 | 2013-2014 | MDN |
| MN27 | Lamberton | 44.24 | -95.30 | 367 | 2013-2014 | MDN |
| MO03 | Ashland Wildlife Area | 38.75 | -92.20 | 257 | 2013-2014 | MDN |
| MT05 | Glacier National Park-Fire Weather Station | 48.51 | -114.00 | 964 | 2013-2014 | MDN |
| NE15 | Mead | 41.15 | -96.49 | 352 | 2013-2014 | MDN |
| NY20 | Huntington Wildlife | 43.97 | -74.22 | 500 | 2013-2014 | MDN |
| NY68 | Biscuit Brook | 41.99 | -74.50 | 634 | 2013-2014 | MDN |
| PA37 | Waynesburg | 39.82 | -80.29 | 452 | 2013-2014 | MDN |
| MI48 | Seney National Wildlife Refuge-Headquarters | 46.29 | -85.95 | 220 | 2013-2014 | MDN |
| SC05 | Cape Romain National Wildlife Refuge | 32.94 | -79.66 | 1 | 2013-2014 | MDN |
| SC03 | Savannah River | 33.25 | -81.65 | 90 | 2013-2014 | MDN |
| PA60 | Valley Forge | 40.12 | -75.88 | 46 | 2013-2014 | MDN |
| PA30 | Erie | 42.16 | -80.11 | 177 | 2013-2014 | MDN |

| Site ID | Site Name | Latitude | Longitude | Elevation | Measurement | Network/ |
|---------|-----------|----------|-----------|-----------|-------------|----------|
| AL03 | Centreville | 32.90 | -87.25 | 135 | 2013-2014 | MDN |
| GA40 | Yorkville | 33.93 | -85.05 | 395 | 2013-2014 | MDN |
| MO46 | Mingo National Wildlife Refuge | 36.97 | -90.14 | 105 | 2013-2014 | MDN |
| KY10 | Mammoth Cave National Park | 37.13 | -86.15 | 236 | 2013-2014 | MDN |
| MS22 | Oak Grove | 30.98 | -88.93 | 100 | 2013-2014 | MDN |
| WI31 | Devil's Lake | 43.44 | -89.68 | 389 | 2013-2014 | MDN |
| PA47 | Millersville | 39.99 | -76.39 | 84 | 2013-2014 | MDN |
| GA33 | Sapelo Island | 31.40 | -81.28 | 3 | 2013-2014 | MDN |
| OK99 | Stilwell | 35.75 | -94.67 | 299 | 2013-2014 | MDN |
| NV02 | Lesperance Ranch | 41.50 | -117.50 | 1388 | 2013-2014 | MDN |
| MD99 | Beltsville | 39.03 | -76.82 | 46 | 2013-2014 | MDN |
| MD08 | Piney Reservoir | 39.71 | -79.01 | 769 | 2013-2014 | MDN |
| NJ30 | New Brunswick | 40.47 | -74.42 | 21 | 2013-2014 | MDN |
| ON07 | Egbert | 44.23 | -79.79 | 196 | 2013-2014 | MDN |
| WI10 | Potawatomi | 45.56 | -88.81 | 570 | 2013-2014 | MDN |
| WA03 | Makah National Fish Hatchery | 48.29 | -124.65 | 6 | 2013-2014 | MDN |
| CA94 | Converse Flats | 34.19 | -116.91 | 1724 | 2013-2014 | MDN |
| CA20 | Yurok Tribe-Requa | 41.56 | -124.09 | 110 | 2013-2014 | MDN |
| OK01 | McGee Creek | 34.32 | -95.89 | 195 | 2013-2014 | MDN |
| OK31 | Copan | 36.91 | -95.88 | 255 | 2013-2014 | MDN |
| SD18 | Eagle Butte | 44.99 | -101.24 | 742 | 2013-2014 | MDN |
| MD00 | Smithsonian Environmental Research Center | 38.89 | -76.56 | 20 | 2013-2014 | MDN |
| FL97 | Everglades-Western Broward County | 26.17 | -80.82 | 4 | 2013-2014 | MDN |
| UT97 | Salt Lake City | 40.71 | -111.96 | 1297 | 2013-2014 | MDN |
| OK04 | Lake Murray | 34.10 | -97.07 | 245 | 2013-2014 | MDN |
| PA52 | Little Pine State Park | 41.36 | -77.36 | 228 | 2013-2014 | MDN |
| KS03 | Reserve | 39.98 | -95.57 | 265 | 2013-2014 | MDN |
| KS24 | Glen Elder State Park | 39.51 | -98.34 | 456 | 2013-2014 | MDN |
| KS99 | Cimarron National Grassland | 37.13 | -101.82 | 1021 | 2013-2014 | MDN |
| OK06 | Wichita Mountains | 34.73 | -98.71 | 492 | 2013-2014 | MDN |
| KS04 | West Mineral | 37.27 | -94.94 | 274 | 2013-2014 | MDN |
| NY43 | Rochester | 43.15 | -77.55 | 136 | 2013-2014 | MDN |
| NY06 | Bronx | 40.87 | -73.88 | 68 | 2013-2014 | MDN |
| MN98 | Blaine | 45.14 | -93.22 | 275 | 2013-2014 | MDN |
| MS12 | Grand Bay NERR | 30.43 | -88.43 | 2 | 2013-2014 | MDN |
| PA21 | Goddard State Park | 41.43 | -80.15 | 385 | 2013-2014 | MDN |
| FL96 | Pensacola | 30.55 | -87.38 | 45 | 2013-2014 | MDN |
| AL19 | Birmingham | 33.55 | -86.81 | 200 | 2013-2014 | MDN |
| DE0008R | Schmücke | 50.65 | 10.77 | 937 | 2013-2014 | EMEP[b] |
| FI0036R | Pallas (Matorova) | 68.00 | 24.24 | 340 | 2013-2014 | EMEP |
| GB0036R | Harwell | 51.57 | -1.32 | 137 | 2013-2014 | EMEP |
| GB0048R | Auchencorth Moss | 55.79 | -3.24 | 260 | 2013-2014 | EMEP |
| NO0001R | Birkenes | 58.38 | 8.25 | 190 | 2013-2014 | EMEP |
| SE0005R | Bredkälen | 63.85 | 15.33 | 404 | 2013-2014 | EMEP |
| SE0011R | Vavihill | 56.02 | 13.15 | 175 | 2013-2014 | EMEP |
| SE0014R | Råö | 57.39 | 11.91 | 5 | 2013-2014 | EMEP |
| NYA | Ny-Ålesund | 78.90 | 11.88 | 12 | 2013-2014 | GMOS[c] |
| MHE | Mace Head | 53.33 | -9.91 | 5 | 2013 | GMOS |
| ISK | Iskrba | 45.56 | 14.86 | 520 | 2013-2014 | GMOS |
| SIS | Sisal | 21.16 | -90.05 | 7 | 2013-2014 | GMOS |
| AMS | Amsterdam Island | -37.80 | 77.55 | 3 | 2013-2014 | GMOS |

| Site ID | Site Name | Latitude | Longitude | Elevation | Measurement | Network/ |
|---------|-----------|----------|-----------|-----------|-------------|----------|
| CGR | Cape Grim | -40.68 | 144.69 | 94 | 2013-2014 | GMOS |
| MCB | Mt. Changbai | 42.41 | 128.11 | 736 | 2011-2014 | China[d] |
| MDM | Mt. Damei | 29.63 | 121.57 | 550 | 2012-2014 | China |
| MLG | Mt. Leigong | 26.39 | 108.20 | 2176 | 2008-2009 | China |
| MAL | Mt. Ailao | 24.53 | 101.11 | 2450 | 2011-2014 | China |
| MWA | Mt. Waliguan | 36.29 | 100.90 | 3816 | 2012-2014 | China |
| BYB | Bayinbuluk | 42.89 | 83.72 | 2500 | 2013-2014 | China |
| PEN | Pengjiayu | 25.63 | 122.07 | 102 | 2009 | Taiwan[e] |
| PR20 | El Verde | 18.32 | -65.82 | 380 | 2015 | MDN |

(a) http://nadp.sws.uiuc.edu/mdn/
(b) http://www.nilu.no/projects/ccc/index.html
(c) Sprovieri et al. (2017)
(d) Fu et al. (2016)
(e) Sheu and Lin (2013)

**Table S2: List of ground stations with observations of Hg(II) surface concentrations used in this study**

| Site ID | Site Name | Latitude | Longitude | Elevation (m.a.s.l.) | Measurement period | Network/ Region |
|---------|-----------|----------|-----------|----------------------|--------------------|-----------------|
| AL19 | Birmingham | 33.55 | -86.81 | 177 | 2009-2012 | AMNet[a] |
| CA48 | Elkhorn Slough | 36.81 | -121.78 | 10 | 2010-2011 | AMNet |
| FL96 | Pensacola | 30.55 | -87.38 | 44 | 2009-2012 | AMNet |
| GA40 | Yorkville | 33.93 | -85.05 | 394 | 2009-2012 | AMNet |
| MD08 | Piney Reservoir | 39.71 | -79.01 | 761 | 2009-2012 | AMNet |
| MD96 | Beltsville_B | 39.03 | -76.82 | 47 | 2009-2012 | AMNet |
| MD97 | Beltsville | 39.03 | -76.82 | 47 | 2009-2012 | AMNet |
| MS12 | Grand Bay NERR | 30.41 | -88.40 | 1 | 2009-2012 | AMNet |
| MS99 | Grand Bay NERR_B | 30.41 | -88.40 | 1 | 2009-2012 | AMNet |
| NH06 | Thompson Farm | 43.11 | -70.95 | 25 | 2009-2011 | AMNet |
| NJ05 | Brigantine | 39.46 | -74.45 | 8 | 2009-2012 | AMNet |
| NS01 | Kejimkujik | 44.43 | -65.20 | 158 | 2009-2012 | AMNet |
| NY06 | New York City | 40.87 | -73.88 | 26 | 2009-2012 | AMNet |
| NY20 | Huntington Wildlife Forest | 43.97 | -74.22 | 502 | 2009-2012 | AMNet |
| NY43 | Rochester | 43.15 | -77.62 | 154 | 2009 | AMNet |
| NY95 | Rochester_B | 43.15 | -77.55 | 154 | 2009-2012 | AMNet |
| OH02 | Athens | 39.31 | -82.12 | 274 | 2009-2012 | AMNet |
| OK99 | Stilwell | 35.75 | -94.67 | 300 | 2009-2012 | AMNet |
| PA13 | Allegheny Portage | 40.46 | -78.56 | 739 | 2009-2012 | AMNet |
| UT96 | Antelope Island | 41.09 | -112.12 | 1285 | 2009-2011 | AMNet |
| UT97 | Salt Lake City | 40.71 | -111.96 | 1099 | 2009-2012 | AMNet |
| VT99 | Underhill | 44.53 | -72.87 | 397 | 2009-2012 | AMNet |
| WI07 | Horicon | 43.46 | -88.62 | 272 | 2009-2012 | AMNet |
| WV99 | Canaan Valley Institute | 39.12 | -79.45 | 985 | 2009-2012 | AMNet |
| AMS | Amsterdam Island | -37.80 | 77.55 | 70 | 2012-13 | GMOS[b] |
| RAO | Råö | 57.39 | 11.91 | 7 | 2012-15 | GMOS[c] |
| LON | Longobucco | 39.39 | 16.61 | 1379 | 2013 | GMOS[d] |
| MAN | Manaus | -2.89 | -59.97 | 110 | 2013 | GMOS[d] |
| WAL | Waldhof | 52.80 | 10.76 | 74 | 2009-2011 | Germany[e] |
| MCH | Mt. Changbai | 42.40 | 128.11 | 740 | 2013-2014 | China[f] |
| MWA | Mt. Waliguan | 36.29 | 100.90 | 3816 | 2007-2008 | China |
| MAL | Mt. Ailao | 24.53 | 101.02 | 2450 | 2011-2012 | China |
| SLA | Shangri-La | 28.02 | 99.73 | 3580 | 2009-2010 | China |
| MYU | Miyun | 40.48 | 116.76 | 220 | 2008-2009 | China |
| MDA | Mt. Damei | 29.63 | 121.57 | 550 | 2011-2013 | China |
| MGO | Mt. Gongga | 29.65 | 102.12 | 1640 | 2005-2007 | China |
| LABS | Lulin Atmospheric Background Station | 23.51 | 120.92 | 2862 | 2006-2007 | Taiwan[g] |
| ALE | Alert | 82.49 | -62.34 | 210 | 2009-2011 | Canada[h] |

(a) http://nadp.sws.uiuc.edu/amn/
(b) Angot et al. (2014)
(c) Wängberg et al. (2016)
(d) Travnikov et al. (2017)
(e) Weigelt et al. (2013)
(f) Fu et al. (2015)
(g) Sheu et al. (2010)

[Figure]

**Figure S1 (a) Simulated and observed Hg wet deposition flux for GMOS and other stations listed in Table S1. (b) Simulated and observed annual volume-weighted mean (VWM) Hg concentration for GMOS and other stations listed in Table S1. The number of stations (N_STA), normalized mean bias (NMB;**

$$\mathrm{NMB} = \sum_i \left(\mathrm{M_i} - \mathrm{O_i}\right) \Big/ \sum_i \mathrm{O_i} \times 100\%$$ **), and FAC2 (percentage of points where** $0.5 \leq \mathrm{M_i}/\mathrm{O_i} \leq 2$ **where** $\mathrm{O_i}$ **and** $\mathrm{M_i}$

**are observed and simulated values, respectively) is included in both panels.**

[Figure]

**Figure S2 Simulated and observed surface Hg(II) concentration for GMOS and other stations listed in Table S1. Note the logarithmic scale on both axes.**

[Figure]

**Figure S3 (a)** Simulated and observed Hg(II) concentrations for aircraft-based campaign over Tullahoma, TN, USA (2012-2013) (Brooks et al., 2013). **(b)**Simulated and observed Hg(II) concentrations for the NOMADSS aircraft-based campaign (2013) (Shah et al., 2016). The number of model-observation pairs in each height bin is shown in panel (a). In panel (b), the number of model-observation pairs in each height bin, and, in parentheses, the number of model-observation pairs where the observations were above the instrument detection limit, are shown.

[revised manuscript text omitted]